# Controllable gliders in a nanomagnetic metamaterial

Arthur Penty [1] ✉, Johannes H. Jensen [1], Ida Breivik [2], Anders Strømberg [2], Erik Folven [2] & Gunnar Tufte [1]

Artificial Spin Ice (ASI) are promising metamaterials for neuromorphic computing, composed of interacting nanomagnets arranged in the plane. Every computing device requires the ability to transform, transmit and store information. While ASI excel at data transformation, their transmission and storage capabilities have been lacking. Here, we take inspiration from Cellular Automata, where information transmission and storage can be realised by the glider, a simple moving structure. Employing an evolutionary algorithm, we discover the snake, a glider in pinwheel ASI. The snake is controlled by a global field protocol, providing precise manipulation of a magnetic texture on the order of 100 nm. We present the snake, both in simulation and experimentally, investigate the mechanism behind its movement and its robustness to disorder. Finally, we demonstrate how the snake can be exploited for computation and memory. The snake enables the integration of information transmission, storage and transformation into one magnetic substrate, unlocking the potential for ultra-low power computing devices.

Artificial Spin Ice (ASI) are metamaterials composed of many nanomagnets arranged on a 2D lattice. The nanomagnets behave as binary artificial spins, which interact through dipolar coupling. By altering the placement and orientation of the nanomagnets, a wide variety of emergent behaviour can be achieved, e.g., long-range ordering[1,2], magnetic monopoles and Dirac strings[3], charge screening[4], and adherence to the ice rules[5,6]. Recently, a new class of field protocols called astroid clocking has been developed that selectively switches nanomagnets within the ensemble[7], offering greater control over the state trajectories of the ASI.

A potential application for ASI is as a substrate for neuromorphic computing[8,9], specifically within a reservoir computing framework[10–15]. Despite promising results[10,12], a major challenge has been a lack of memory capacity[11,13], i.e., a linear correlation between the current reservoir state and past input values[16]. To remedy the low memory capacity in ASI reservoirs, external support circuitries, such as delay line memories have been required[17]. External memories add significant overhead and reduce efficiency[18], since the reservoir state must be constantly read out and stored. There appears to be a fundamental computational property which has thus far eluded ASI computing approaches.

Fundamentally, any computing substrate must support the transmission, storage and transformation of information[19]. Here we take inspiration from the Cellular Automaton (CA), a simple lattice-based model of distributed computing. In one of the most famous 2D CAs, Conway's Game of Life (GoL)[20], information transmission can be realised by the glider: a small 5-cell structure that can travel indefinitely through the CA, until it collides with another structure. Gliders and other glider-like structures can be used to transmit signals and form the basis of a CA computing model[21]. Additionally, the reliable movement of the glider can be used as a basis for memory, since information can flow through the system without degrading. The concept of a glider can be generalised beyond GoL, to any structure that can move through a substrate while maintaining its form, i.e., translation[22]. While freely moving monopoles found in ASI[3,23–25] may seem a potential candidate for gliders, they leave a trail of reversed magnets (Dirac strings), impeding subsequent monopoles. Our goal is not to implement GoL in ASI, but to search for configurations where glider-like behaviour can be found.

[1]Department of Computer Science, Norwegian University of Science and Technology, Trondheim, Norway. [2]Department of Electronic Systems, Norwegian University of Science and Technology, Trondheim, Norway. ✉e-mail: arthur.penty@ntnu.no

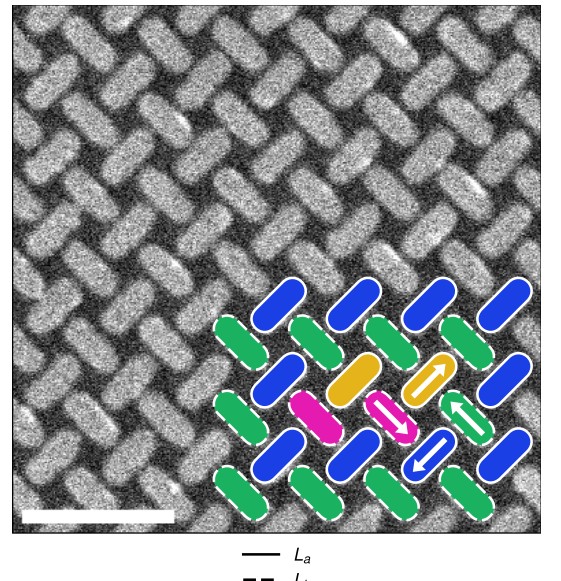

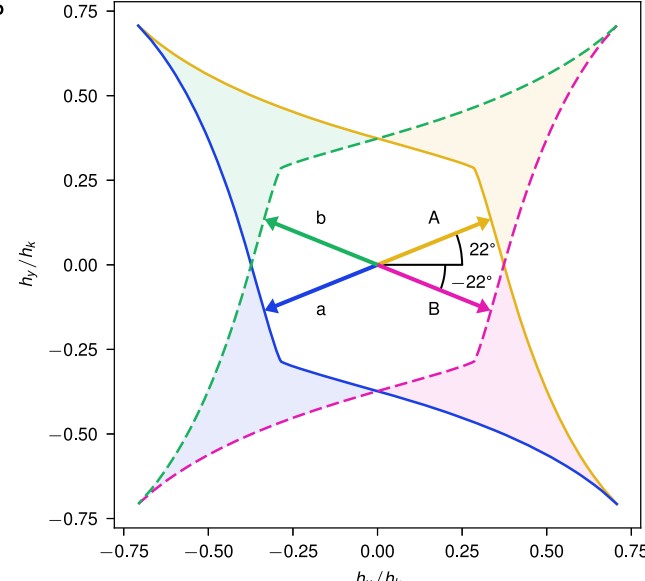

$L_a$ ———
$L_b$ - - -

**Fig. 1 | Astroid clocking of pinwheel ASI. a** A scanning electron microscopy image of a pinwheel ASI. The inset in the lower-right corner illustrates magnetisation with colours corresponding to magnetisation directions, as indicated by the white arrows. The inset shows a small rightwards (orange-pink) domain surrounded by a leftwards (blue-green) background domain. The scale bar is 500 nm. **b** The two switching astroids for the nanomagnets in sublattices $L_a$ (solid outline) and $L_b$ (dashed outline), where $h_k$ is the coercive field strength along the hard axis, and $h_x$ and $h_y$ are the horizontal and vertical components of an external field. The astroid boundaries indicate the required magnetic field to switch a magnet in the corresponding sublattice. Astroid edges are coloured according to the magnet state promoted when crossing the edge. Similarly, the shaded areas indicate regions where selective switching occurs, promoting a single magnet colour from a single sublattice. In the centre of the astroids, the two bipolar clocks A and B are depicted at +22° and −22°, respectively.

Beyond computing, a magnetic glider would provide a means to manipulate a magnetic texture with high precision. Magnetic gliders could be leveraged to precisely control magnetic phenomena in other systems. For example, creating channels for spin-waves in reconfigurable magnonic crystals[26] or guiding the movement of magnetic nanoparticles[27]. The control offered by a magnetic glider could unlock a range of possible research directions and future devices.

Can gliders be found in an ASI? Here, we show that the answer is a definite yes. Using an Evolutionary Algorithm (EA), we search for gliders in pinwheel ASI and present the simplest glider structure discovered, which we term the snake. The snake glider can move either left or right, with the direction dictated by the orientation of its structure. We verify the snake both in simulation and in experiment, and analyse the mechanism behind its movement. Next, we investigate the robustness of the snake, where we find evidence of self-correction and graceful degradation. Finally, we demonstrate how the snake can be exploited for computation, integrating the transformation, transmission and storage of information into a single magnetic substrate.

## Results

Pinwheel ASI, depicted in Fig. 1a, is composed of two interleaved square sublattices $L_a$ and $L_b$, whose nanomagnets are rotated +45° or −45°, respectively. In this work, we refer to the sublattice and magnetisation of the nanomagnets by their colour (blue/orange for $L_a$ and green/pink for $L_b$), as indicated by the coloured inset in Fig. 1a. Pinwheel ASI exhibits long-range ferromagnetic order, supporting large domains of coherent magnetisation. We consider a 50 × 50 array with a background domain of leftwards magnetisation (blue-green). In the centre of the array, we initialise some magnetic structure with magnetisation in the opposite direction (orange-pink), which is considered the initial state of the ASI.

To drive the pinwheel ASI we employ astroid clocking[7]. Figure 1b shows the switching astroids for the nanomagnets in the two sublattices $L_a$ (solid outline) and $L_b$ (dashed outline). A nanomagnet will switch (reverse magnetisation) if subject to a magnetic field that crosses the astroid boundary. The shaded areas in the astroids indicate fields that selectively switch magnets from a single sublattice whose magnetisation is oppositely aligned to the field.

We define two clocking directions $A$ and $B$ along the +22° and −22° axes respectively, as shown in Fig. 1b. $A$ and $B$ denote the positive (rightwards) clock field, while $a$ and $b$ refer to the negative (leftwards) clock fields. $A$ and $a$ may selectively switch magnets from sublattice $L_a$, while $B$ and $b$ selectively switch magnets from $L_b$. For example, applying the $A$ field will selectively switch blue magnets to the orange state.

Previous work[7] has demonstrated that $AB$ clocking (pulsing the $A$ and $B$ fields in an alternating fashion) will result in step-wise growth of rightwards (orange-pink) domains. Similarly, $ab$ clocking results in growth of blue-green domains and consequently reversal of orange-pink domains. For a domain contained within a larger outer domain, reversal is generally faster than growth, when the field strengths are equal. Typically, domains are observed to grow or shrink, but not to move through the ASI, making this behaviour unsuitable for information transmission.

An orange-pink magnetic domain grows under positive fields $A$ and $B$, while shrinking under negative fields $a$ and $b$. A glider moves while retaining its shape, and thus requires a balance of growing and shrinking. Hence, we decouple the strength of the positive and negative clock fields, allowing for the possibility of a *wonky* clock protocol. We define $H^+$ as the strength of $A$ and $B$, and $H^-$ as the strength $a$ and $b$. We fix the clock protocol to $aAbB$, which can cause both growth and reversal within a clock cycle. Glider discovery thus requires both an initial glider structure, and suitable field strengths.

We use an EA, a bio-inspired optimisation algorithm, to efficiently search the large parameter space for regions that may support glider-like behaviour. The EA is given the task of finding an initial ASI state along with suitable values for $H^+$ and $H^-$. To evaluate a potential glider, we simulate its trajectory under the clock protocol using flatspin, the large-scale ASI simulator[28].

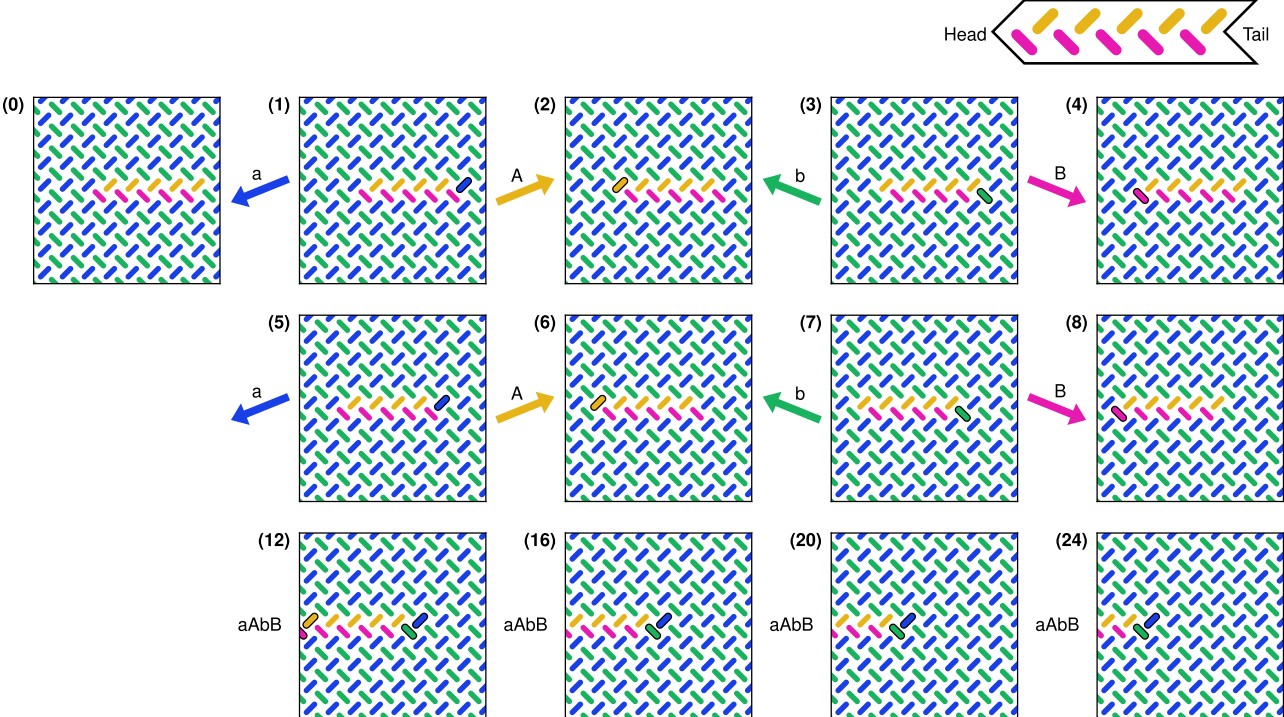

**Fig. 2 | Snake glider in pinwheel ASI.** The inset in the top-right shows the snake glider and defines its head and tail. The snapshots (0-24) shows a zoomed-in view of a 50 × 50 system, at different points during the *aAbB* clock protocol. (0) shows the initial state of the snake, an elongated orange-pink domain in the centre of a blue-green background domain. (1–8) show the state of the snake during *aAbB* clocking, where magnets that change state between snapshots are highlighted by a solid black outline. The field arrow to the left of each snapshot indicates the clock field which precedes it. The snake moves leftwards through alternate shrinking (*a* and *b*) and growing (*A* and *B*) steps. The bottom row continues the series, showing only snapshots after a complete clock cycle (four clock fields are applied between each snapshot). Note that, despite the snake leaving the viewing window, it is still far from the edge of the ASI.

EAs typically require a fitness function to define the behaviour they are to target. There are many ways one could attempt to quantify glider-like behaviour. Here, we take a phenomenological approach, matching abstract properties of the ASI trajectory to those expected of a glider. Conceptually, our fitness function consists of two parts, the first part attempts to balance the growth and shrinking of the initial domain over time, minimising the difference between the number of magnets joining the domain versus those leaving the domain. While this captures the idea that movement is the balance of growing and shrinking, it also allows for other possibilities; namely, frozen or oscillatory behaviours. To steer the EA away from these behaviours we introduce a second part to our fitness function, a measure of the transient length of the trajectories, i.e., the number of states the ASI passes through before a state is revisited. By penalising short transient lengths, we make the frozen and oscillatory behaviours undesirable to the EA. Defining the fitness function in this abstract way results in a hands-off approach where we do not bias the EA towards any pre-conceived ideas of how a glider should function in ASI, such as its direction of travel or the mechanisms underlying its movement.

Another consideration is the need for gradients in the fitness function. EAs perform best on fitness functions where small incremental improvements are rewarded, rather than those with a hard cut-off between desirable and undesirable behaviour[29]. Our construction of a fitness function lends itself nicely to incremental improvements, both the balance between growing and shrinking, and the trajectory length, can be improved in small repeated steps.

For further details on the EA and fitness function, see Methods.

## The snake glider
The EA discovered multiple interesting structures with glider-like properties (see Supplementary information for other glider discoveries). Here we focus on the snake, shown at the top of Fig. 2, the simplest glider discovered. The shape of the snake glider resembles an arrow, with a pointed head and split tail. The snake achieves a perfect score in the fitness function, corresponding to an exact match of glider behaviour. Figure 2 (0) shows the initial state of the snake, consisting of a thin, elongated domain of orange and pink magnets. Figure 2 (1–8) show the movement of the snake, where the snake is fully translated one step leftwards every four field applications (one full clock cycle). Figure 2 (12, 16, 20, 24) show a continuation of the series, each a full clock cycle apart. Further application of the clocking protocol will continue to translate the snake leftwards, until it reaches the edge of the ASI. The behaviour of the snake resembles that of the 1D shift register constructed by Nomura et al.[30], though here it is able to function on a 2D lattice. A video of the simulated snake glider is available in Supplementary Movie 1.

The example in Fig. 2 shows a leftwards-moving snake, but surprisingly, if the snake is inverted with respect to the sublattices (pink magnets on the top, orange on the bottom), the snake then proceeds rightwards. No changes to the clock protocol are needed for this reversal in direction, it is purely a function of the magnetic state. Consequently, two snakes can inhabit the same ASI, experience the same clock protocol, but move in opposite directions, due only to their structure (see Supplementary information and Supplementary Movie 2).

We can understand the underlying mechanism of the snake glider by considering the effect of each individual field in the clock protocol. Figure 3 provides a schematic rule set of the snake movement. We see the $A$ and $B$ fields grow the head on the $L_a$ and $L_b$ sublattices, while leaving the tail unchanged. Similarly, the $a$ and $b$ fields shrink the tail on the $L_a$ and $L_b$ sublattices, leaving the head unchanged. When applied sequentially, the $a$ field switches the orange magnet at the tail,

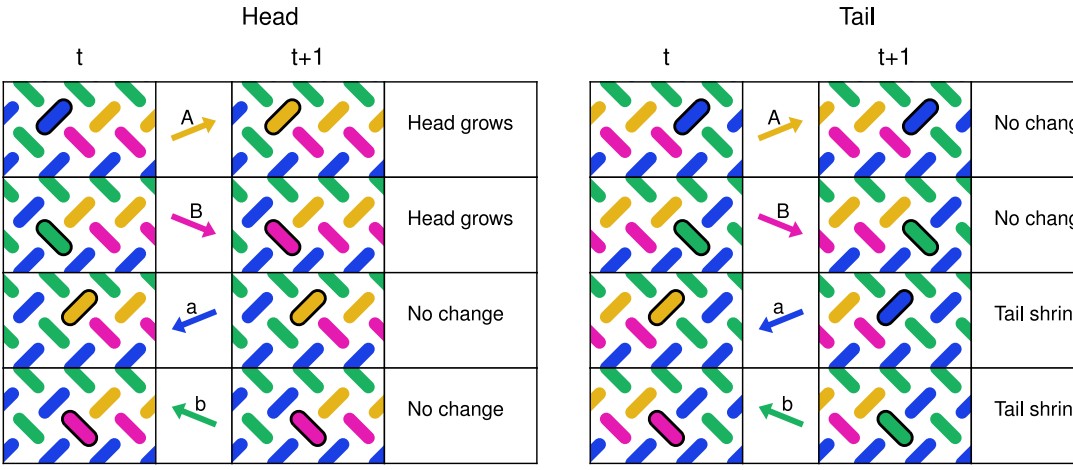

**Fig. 3 | A schematic rule set, illustrating how the head and tail of the left-moving snake respond differently to the clocking fields.** For each field we consider only the relevant head and tail configurations, i.e., configurations where the edgemost magnet is on the sublattice targeted by the field. In each image we highlight this magnet of interest with an outline.

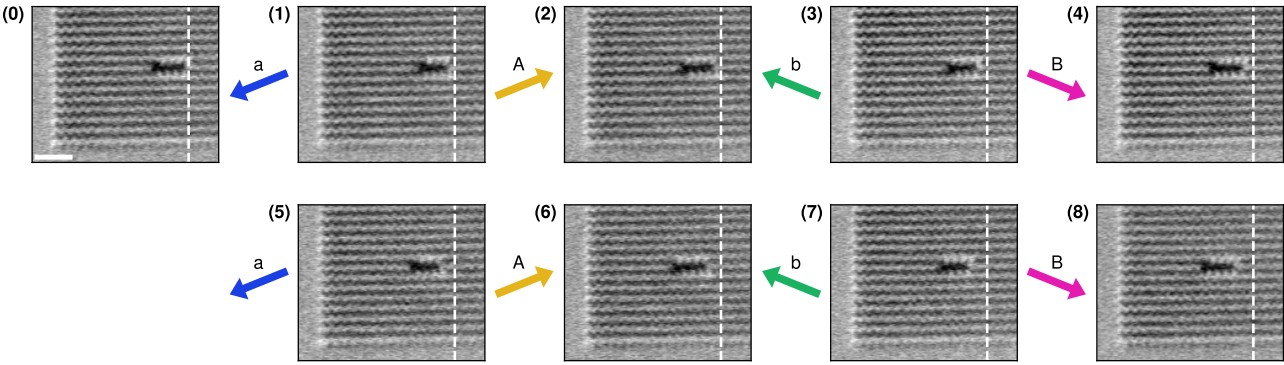

**Fig. 4 | Experimental realisation of the snake glider in pinwheel ASI.** Each MFM-micrograph shows a part of a 100 × 100 pinwheel ASI, after application of a clock field in the *aAbB* clock protocol. All MFM-micrographs are taken at zero field. (0) shows the initial state of the snake, created by field-assisted writing using an MFM tip. The initial state is followed by a series of applied fields. (1–8) show the movement of the snake during *aAbB* clocking. The tail position of the initial snake is highlighted by the white dashed line. The scale bar in (0) is 1 μm.

and unlocks the neighbouring pink magnet, which is in turn switched by the *b* field. A similar unlocking occurs for growth at the head. Due to this unlocking mechanism, the clock fields can in fact be applied in any order, provided each is applied exactly once within a clock cycle. Furthermore, the length of the snake can be modified by applying only the positive fields *A* and *B*, causing the head to grow while the tail remains fixed. Likewise, repeated cycling of the *a* and *b* fields causes the tail to shrink while the head remains fixed. The snake can then be driven by the full clock protocol again and will retain its glider properties at its new size. The smallest functional snake is of length one (one magnet on each sublattice). As the head and tail function independently, there is no upper bound on the length of the snake, provided it does not reach the edge of the ASI.

For verification of the snake glider, we reproduced the behaviour in the micromagnetic simulator MuMax3[31]. These results are shown in the Supplementary information and corroborate the results of flatspin. A video of the MuMax3 simulation is available in Supplementary Movie 3.

**Experimental demonstration**

Next, we demonstrate snake movement experimentally. The initial snake state is written onto a polarised ASI using a magnetic force microscopy (MFM) tip and a small bias field[32,33]. Then the *aAbB*

protocol is applied using an in-plane quadrupole vector magnet. The ASI is imaged using MFM between each applied field. See Methods for further details.

Figure 4 shows MFM-micrographs of the snake after each applied clock field. The snake is visible in black, against the striped background of the polarised outer domain (blue-green in Fig. 2). In the MFM-micrographs, the starting position of the tail is marked with a dashed white line. Figure 4 (0) shows the initial state of the snake. In Fig. 4 (1–8), we move the snake for two clock cycles, with behaviour identical to our simulations. Comparing MFM-micrographs (0) and (8), the snake has moved leftwards by two lattice spacings, corresponding to ~495 nm.

After further clocking, the movement broke down due to the erroneous switching of some magnets close to the tail. These magnets likely have lower coercive fields, causing the snake to change shape and lose its glider behaviour. Despite these tail defects, the snake continues to move for two more clock cycles. A video of the full experimental series is available in Supplementary Movie 4.

These results show the snake movement is experimentally feasible despite the inevitable variations in coercive fields. We attribute the variation in switching fields primarily to fabrication defects in the ASI. In the Robustness section, we further examine the effect of such disorder.

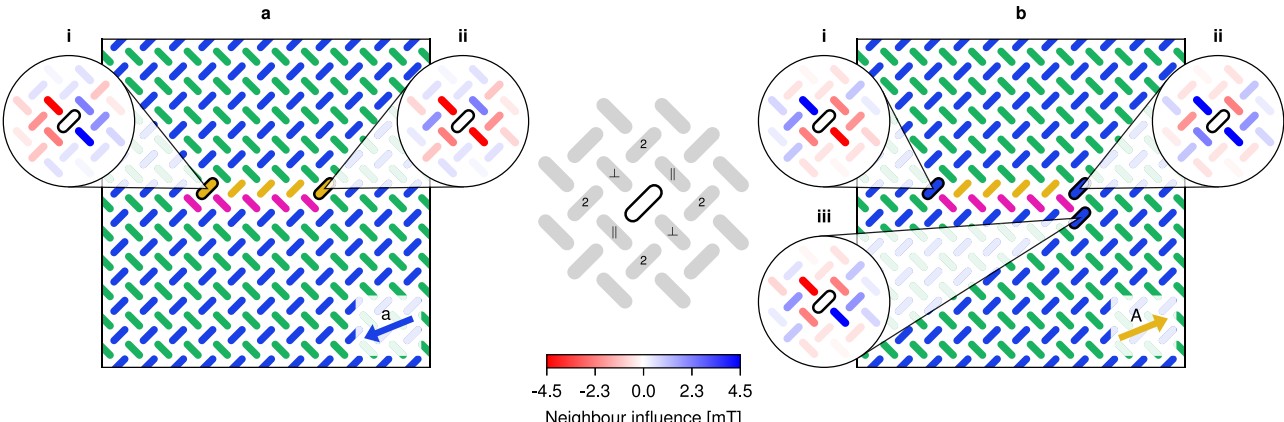

**Fig. 5 | Analysis of neighbour influence in the snake.** Neighbour influence during (**a**) shrinking and (**b**) growth. The five insets illustrate how the highlighted magnet is influenced by its neighbours through their dipolar fields. The influence of a neighbour acting on a magnet is given by the resulting change in the magnet's proximity to the switching astroid when considering the neighbour's dipolar field. Positive influence values (blue) indicate an increased distance (stabilising) and negative values (red) indicate a decreased distance (destabilising). The centre legend shows the parallel (∥) and perpendicular (⊥) nearest neighbours, as well as the second nearest neighbours (**2**).

## Analysis

While the EA discovered the snake at specific field strengths, it is able to function over a range of strengths. In general, we find that the valid $H^-$ values tend to be lesser than those for $H^+$ (see Robustness). Experimentally, applying a negative clock field with strength equal to the positive clock field switches more than a single tail nanomagnet, causing the snake state to disappear (see Supplementary information). Intuitively, this can be understood by considering the two domains being grown. The positive fields grow the smaller, inner domain (the snake), whereas the negative fields cause the larger, outer domain to collapse inwards. The larger domain grows faster due to the morphology of the domain boundary, i.e., it envelopes the inner domain[7]. It follows then, that the field for growing the larger domain ($H^-$) should be lesser in order to counterbalance the system's inherent bias towards reversal.

Although the shape of the gliders is reminiscent of Dirac strings connecting emergent monopoles in ASI[34,35], they are fundamentally different. In the pinwheel geometry, there are no well-defined vertices where the figuratively charged ends of neighbouring nanomagnets combine directly to create emergent magnetic point charges. While it is possible to view the tilted nanomagnets as contributing a partial charge to each pinwheel vertex, this perspective yields limited understanding of the glider behaviour (see Supplementary information). The switching astroids must be taken into account to explain what drives the snake's motion.

To understand the mechanism of the snake's motion, we analyse the effect of each neighbour magnet under an applied clock field. In Fig. 5, the five insets illustrate how the highlighted magnet is influenced by its neighbours through their dipolar fields. Here, the influence of a neighbour on a magnet is given by the resulting change in the magnet's proximity to the switching astroid when considering the neighbour's dipolar field. Neighbour magnets with positive influence values are coloured blue, indicating an increased distance and thus a stabilising interaction. Negative influence values are coloured red, indicating a decreased distance and are thus destabilising. The centre legend of Fig. 5 depicts the local neighbourhood of a magnet. We denote the four nearest neighbours of a magnet as $NN$, and the four next-nearest neighbours as $NNN$. Additionally, we distinguish between $NN_∥$ and $NN_⊥$, corresponding to the $NN$s that lie parallel or perpendicular to the magnet's principal (easy) axis. As can be seen, the $NN_⊥$ magnets exert the strongest influence, changing the proximity to the astroid by ≈ 4.5 mT, followed by the $NN_∥$ magnets with ≈ 2.2 mT.

Growth occurs during the positive fields $A$ and $B$, which we find are sufficiently strong to switch the entire sublattice, in the absence of dipolar fields. Hence, during growth, dipolar interactions have a net stabilising effect, preventing most magnets from switching. In contrast, the negative fields $a$ and $b$ responsible for shrinking, are not strong enough to flip any magnets alone. In this case, switching is mediated by dipolar interactions that have a net destabilising effect, causing some magnets to switch.

Figure 5a depicts the snake under the $a$ field, which effectively shrinks the tail of the snake by switching the rightmost orange magnet. The insets for the two highlighted magnets at the head and tail reveal why only the tail magnet switches. For the head magnet (Fig. 5ai), the influence from the four $NN$ magnets effectively cancels out. (The situation is similar for the orange magnets at the top of the snake). Even though three of the four $NNN$ magnets are in a weakly stabilising configuration, the stability can be attributed primarily to the weakness of the $a$ field, where no switching occurs in the absence of strong destabilising interactions. However, for the tail magnet (Fig. 5aii), there are indeed strong destabilising interactions from its two $NN_⊥$ magnets (bright red).

Figure 5b depicts the situation during growth of the head of the snake, where only the left-most highlighted blue magnet will flip under the $A$ field. Now there are three relevant cases that could switch, namely the blue magnets at the head, below and at the tail of the snake. As can be seen in the inset Fig. 5bi, the head magnet loses stability when the $NN_⊥$ interactions cancel out, and is destabilised even further by its $NN_∥$ neighbours. Meanwhile, the tail magnet (Fig. 5bii) is strongly stabilised by its $NN_⊥$ neighbours. Of the blue magnets beneath the snake, the least stable is the rightmost one immediately below the tail (Fig. 5biii). Note that the $NN$ interactions for this magnet are similar to those of the head magnet, however the $NNN$ interactions are slightly more stabilising. The influence maps for the other blue magnets below the snake are similar, but more stable due to their pink $NN_∥$ neighbour.

For the other sublattice ($L_b$) and corresponding fields ($B$ and $b$), the above analysis is identical (see Supplementary information).

Our analysis above shows that, under the applied clocking fields, the $NN_⊥$ magnets exert the strongest influence on switching (with strongest red or blue colour). This may come as a surprise, since the dipolar fields from these neighbours are oriented perpendicular to the easy axis of the centre magnet. However, due to the sloped edges of the switching astroid (Fig. 1b), a perpendicular field can help push a magnet outside the astroid edge. Hence, the clocked dynamics of the snake can not be understood in terms of dipolar energy alone (which

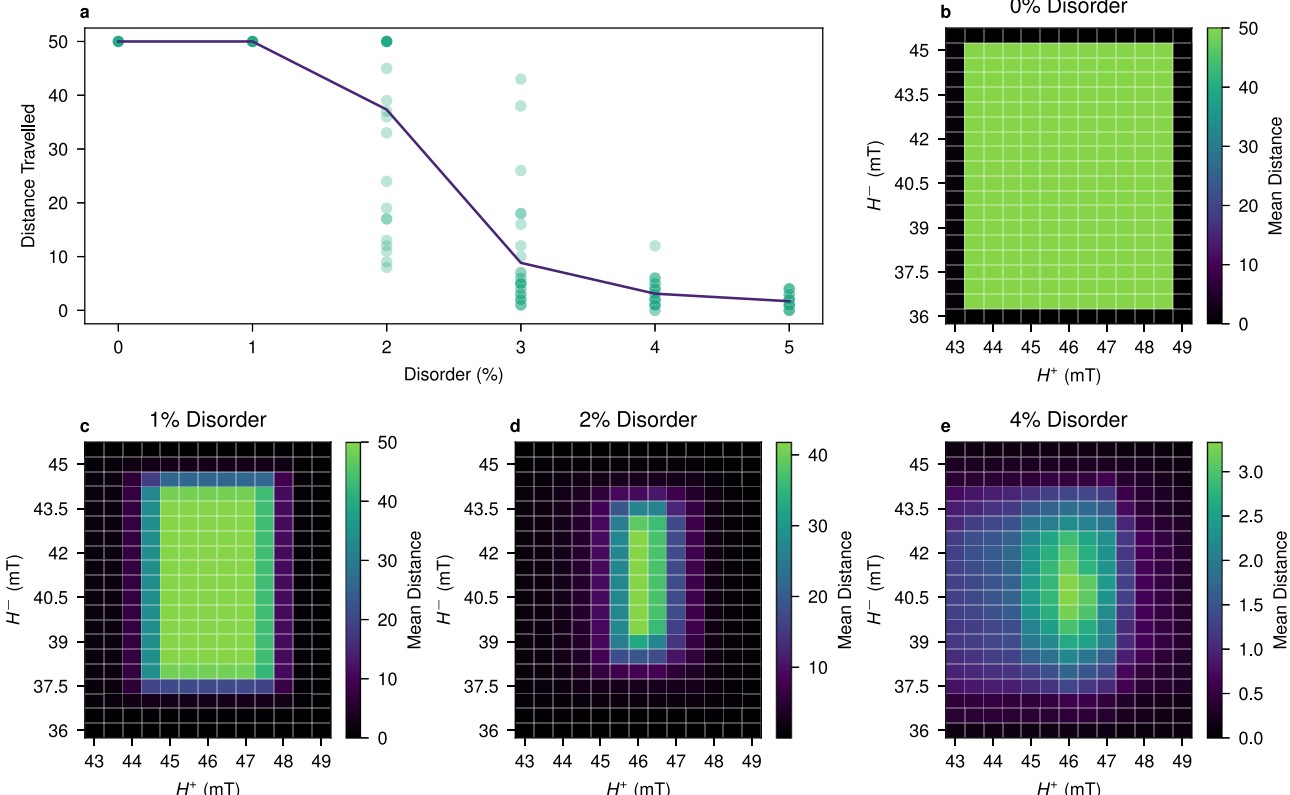

**Fig. 6 | The robustness of the snake to quenched disorder. a** shows distances travelled by a snake glider over 30 different disorder samples for each level of disorder, where $H^+ = 46$ mT and $H^- = 40.5$ mT. Each data point (green dot) represents a single simulation run. The mean distance travelled at each disorder level is indicated by the purple line. **b–e** show heat maps of the mean distance travelled by snake gliders, for disorder values of 0% (no disorder), 1%, 2% and 4%. Each cell represents the snake distance for the pair ($H^+$, $H^-$), averaged over the 30 samples.

only considers the parallel component of the dipolar field). See also the astroid cluster plots in the Supplementary information, where the locations of each magnet within the astroid are shown.

The apparent leftwards movement is a result of strong interactions from the $NN_\perp$ magnets of the tail magnet, and somewhat weaker interactions from the $NN_\parallel$ magnets of the head magnet. In other words, the direction of movement is a result of the pointy shape of the snake's head and the split shape of its tail. If the snake is inverted with respect to the sublattices, the pointy head will be on the right side. As a result, the inverted snake will move towards the right.

## Robustness

Next, we investigate the robustness of the snake to quenched disorder. Specifically, we consider the quenched disorder manifest as a variance in the switching thresholds, arising from the fabrication process. In simulation we introduce disorder by adding noise to the switching threshold of each magnet (see Methods). For a concrete measure of robustness we consider the average distance a snake travels before failure. Snake gliders were simulated under a sequence of 50 clock cycles for various levels of disorder (0–5%) and for different field strength pairs ($H^+$, $H^-$), repeated 30 times for each disorder level.

Figure 6a shows the effect of different disorder levels for the field strengths $H^+ = 46$ mT and $H^- = 40.5$ mT. These fields were chosen as the values that performed best at 2% disorder, and were not significantly outperformed at any other disorder level. As expected, at 0% disorder the snakes function perfectly, and this behaviour is maintained at 1% disorder. At 2% disorder, however, the mean distance travelled drops to $\approx$ 73% of the maximum distance of 50. Of the 30 samples of 2% disorder, 16 showed perfect behaviour. Beyond the 2% disorder level, a significant drop off in mean distance is observed.

Figure 6b-e present the expected distance travelled by a snake glider for different field strengths. Figure 6b shows the ideal system, displaying the full range of field strengths which can sustain the snake glider. As disorder increases (Fig. 6c–e), not only does the best mean distance decrease, but also the optimal range of field values shrinks. Consequently, higher disorder levels require greater precision in the external field strengths.

In simulations where the glider does not achieve the maximum distance, three different modes of failure are observed: explosion, where the snake expands uncontrollably; implosion, where the snake shrinks to nothing; and frozen, where either the head or tail of the snake cease to move. There is a clear propensity for different field regions to prefer different failure modes. Intuitively, explosion failures are most prevalent at high $H^+$. High $H^-$ lead to more implosion failures, whilst freezing is most common when one or both the fields are too low. Interestingly, we notice the best performing field-pair for each disorder level tends to produce implosion or frozen failures and rarely explosions. This can be seen as a form of graceful degradation, i.e., when the snake fails the error is locally contained and does not spread to the rest of the system.

Experimentally, we investigate robustness by initialising groups of snakes and applying clock cycles until the snakes explode, implode or freeze (see Methods). Figure 7 shows MFM-micrographs of six snakes under eight $aAbB$ clock cycles. As can be seen, the topmost snake moves for all eight clock cycles, travelling around 1.98 $\mu$m in total. Eight steps of movement was the furthest distance travelled by any snake experimentally. We also see instances of all three failure modes: exploding (fifth snake from the top), imploding (second snake) and freezing (third, fourth and sixth snake). Interestingly the exploding snake produces an anti-ferromagnetic

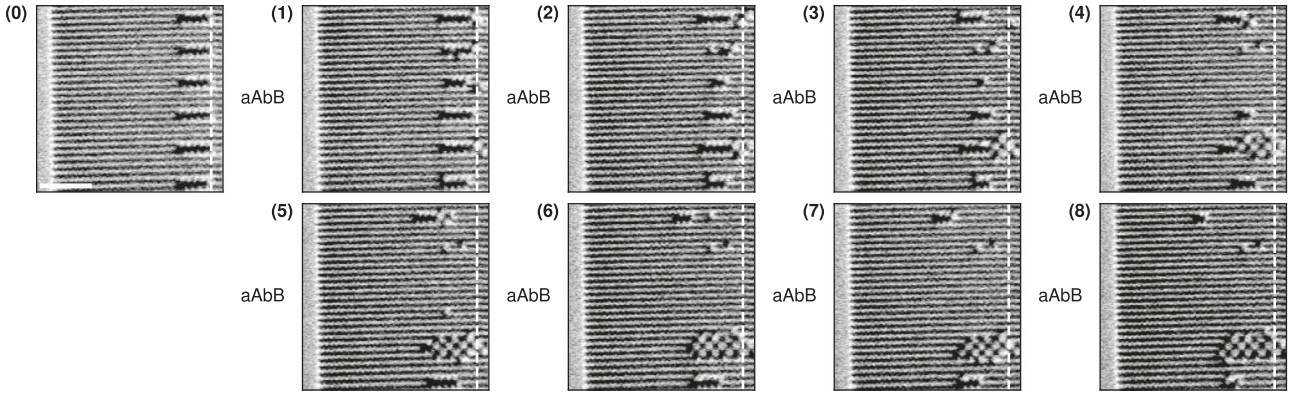

**Fig. 7 | Experimental demonstration of snake robustness. 0** shows the six initialised snakes, one on every fifth row of the pinwheel ASI. The starting tail position of the snakes is marked by the white dashed line. The scale bar is $2\,\mu m$. **1–8** show the snakes after a full cycle of the *aAbB* field protocol.

domain, an unanticipated behaviour in the inherently ferromagnetic pinwheel ASI geometry[1].

While the top snake of Fig. 7 is able to travel eight steps, there are intermediate steps in which it is not a perfect snake. We see in Fig. 7(4), there are some defects present around the snake's tail, but a few steps later, in Fig. 7(6) the snake is again a perfect snake. This demonstrates the snake's ability to self-correct, providing some resilience to disorder.

In addition to the group of snakes shown in Fig. 7, more groups of snakes can be found in Supplementary Movies 5 & 6. We also carried out a repeat for one of the groups of snakes (marked in blue in Supplementary Movie 5), where the snakes were re-initialised in the same positions and clocked with the same field strengths. The corresponding snakes break in similar ways, suggesting that the disorder is in fact quenched, and likely due to fabrication defects. Supplementary Movie 7 shows four series of snakes initialised on the same rows of the pinwheel ASI, but clocked with different $H^+$. As in simulation, when the $H^+$ field becomes too strong, explosion failures are more prevalent.

Comparing to Fig. 6a, it is likely that the experimental disorder is at least 4%, given that the greatest distance travelled observed experimentally was eight steps. The presence of all three failure modes under the same field strengths indicates that the fields are too strong for some snakes (explosion or implosion) and too weak for others (freezing). Hence, at the observed level of disorder, no field strengths can support translation of all snakes simultaneously.

The amount of quenched disorder in the experimental system could be reduced through optimisation of the fabrication process (see Methods). In particular, we believe edge roughness plays a significant role in the observed variation in coercive fields. Hence, greater consistency in nanomagnet morphology should improve disorder.

### Towards computing

As aforementioned, any computing substrate must possess the ability to transmit, store and transform information. The snake's ability to move through an ASI while maintaining its form provides a mechanism for information transfer and storage. Information can be encoded in the snake, e.g., length, position or direction. Transmission is realised in the movement of the snake, while storage results from its preservation of shape. However, information transformation occurs when the snake's form is altered, such as reacting to some external stimuli, or interacting with another magnetic structure in the ASI. Here we consider two potential computing regimes realised in snake gliders. In the first regime, the snakes provides pure transmission and storage. The second regime adds information transformation through snake collisions.

In the first regime, one or more input bits are clocked from one edge of the ASI to the other (Supplementary Movies 8 & 9). In this way,

the snakes realise a shift register embedded in the 2D fabric of the ASI. Due to the importance of *NNN* interactions, a minimum snake length of two is needed to reliably encode a bit. Multiple bits can be encoded in parallel, provided there is sufficient vertical spacing to prevent snake interference. Under this encoding, the memory capacity (MC[16]) is bounded by the width of the ASI. The MC for our snake-based shift register is $MC = W/2 - 2$, where $W$ is the width of the ASI (see Methods). Consequently, for the $50 \times 50$ ASI, $MC = 23$, scaling linearly with the width of the array. As a comparison, the vortex states in width-modified pinwheel[13] support $MC \approx 5$, while $MC \approx 3.5$ was reported for a kagome system[11]. Here, the snake's ability to move while maintaining its form is exploited for a MC that vastly outperforms other proposed mechanisms in ASI.

In the second regime, snake collisions provide a means for information transformation. Snake collisions were simulated under various field strengths $H^+$ and $H^-$ (Supplementary Movie 10). Figure 8 shows the outcome of three collisions of interest, displaying a diverse response to the different field strengths. The top row of Fig. 8 shows two snakes colliding and annihilating, leaving nothing behind. In the second row, the collision cancels the rightwards-moving snake, while the leftwards-moving snake continues to propagate but at a reduced length. In the bottom row, the two snakes collide to form a stable unmoving structure, transforming the moving signal into a non-volatile form for persistent storage. In these examples, the leftwards-moving snake is always one row above the other. Inverting this relationship provides even more possible interactions. Snakes can also interact without direct contact. When the snakes meet with a row of magnets between them, another set of field strength dependent interactions occurs (Supplementary Movie 11).

Though the snake's movement is confined to a single dimension, the results here show it is able to interact with other structures, from both above and below. This allows for the utilisation of the second dimension of the ASI, making the substrate fundamentally more powerful than 1D magnetic structures such as shift registers[30] or racetrack memories[36].

The examples given here are for a disorder free system. In experiment, the interactions observed would be affected by disorder. While disorder reduces the transmission, it brings a variety of transformations, such as the previously observed failure modes (see Robustness). A rich pool of transformations is vital for RC, where the inhomogeneity provided by disorder could be beneficial.

### Discussion

To the best of our knowledge, the snake glider is the first example of the precise and controlled translation of a domain in ASI. In the snake glider we have deterministic, local control using only global clock fields, providing a powerful and practical technique for future

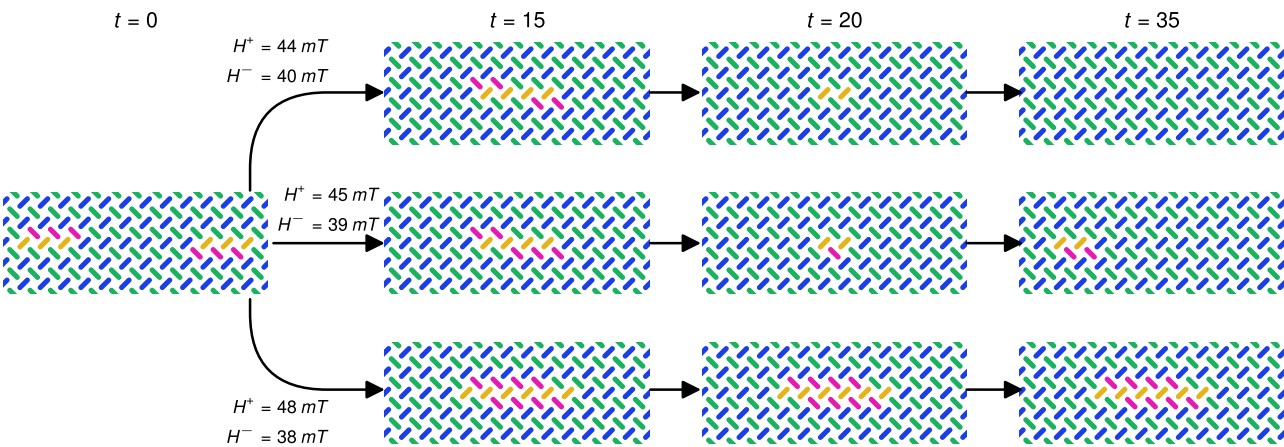

**Fig. 8 | The collision of two snakes.** Each branch shows the resulting collision for a different pair of field strengths ($H^+$, $H^-$). Within each branch the field strengths remain constant.

nanomagnetic devices. The snake enables new directions in controlling other magnetic phenomena, such as creating channels for spin-waves in reconfigurable magnonic crystals[26] or guiding the movement of magnetic nanoparticles[27]. In a practical device, the snake could be nucleated by various means such as all-optical switching[37] or current-controlled writing[38].

The performance of such a device would be affected by disorder. Our analysis indicates that a modest reduction in fabrication defects would significantly improve reliability. Even at high levels of disorder, we see signs of graceful degradation and self-correcting behaviour. Additionally, larger or more complex glider structures may offer greater robustness. Reducing disorder and increasing glider robustness are two complementary paths towards more reliable operation.

Reservoir computing in ASI has so far been limited by poor memory capacity, which we attribute to a lack of information transmission and storage capability. By utilising the snake glider as an information carrier, these capabilities can now be realised. Using the snakes, memory capacity scales linearly with system size, providing a path towards memory-intensive tasks. Consequently, ASI reservoirs can tackle temporal computing tasks directly, without the need for costly peripheral aids.

Going beyond, the snake could serve as the information carrier between larger complex systems, just as the glider in Game of Life based computers. Snake-like structures could be employed to support communication between components in magnetic computing devices. Computational units could be realised directly in the ASI substrate as other kinds of magnetic structures, towards a kind of reconfigurable magnetic device. Integrating the transformation, storage and transmission of data onto the same substrate, eliminates the need to translate information between mediums, unlocking significant efficiency gains. With the discovery of a magnetic glider, ASI now provides all the fundamental requirements for an all-magnetic computing chip.

## Methods
### Evolutionary algorithm
The EA begins with a random population of individuals (solutions). As the algorithm progresses, the individuals are assessed through simulation. The better performing individuals are retained, with a chance of variation (mutation or recombination). Thus, through a process akin to natural evolution, the population undergoes iterative improvement. The EA was run with a population size of 100 for 100 generations.

Individuals in the EA specify two field strengths $H^+$ and $H^-$, and an initial state. The EA has control of the initial state of a roughly-square patch of 128 magnets at the centre of an ASI. The initial state is represented as a one dimensional list of real values, with each element

corresponding to one of the magnets in the centre square region of the ASI. In close analogy to CA theory, we define the orange or pink spin state as *on*, and the blue or green as *off*. A value of greater than 0.5 in the initial state list, causes the corresponding magnet to be initialised in the *on* state (polarised rightwards), all other magnets begin in the *off* state.

To evaluate an individual we simulate its trajectory under the *aAbB* clock protocol using flatspin[28]. The trajectory is the series of ASI states, assembled by sequentially applying the fields of the protocol and recording the resulting spin state of the full ensemble. We define the following fitness function on the resulting trajectory, to be minimised by the EA:

$$\left( \sum_{t=t_0}^{T} |a_t - a_{t_0}| \right) + k \times (N - n_u), \qquad (1)$$

where $a_t$ is the number of magnets in the *on* state at time $t$, $k$ is the penalty factor, $n_u$ is the number of unique ASI states in the period $t \in [t_0, T]$ and $N$ is the length of the trajectory ($T - t_0$).

In minimising this function, the EA rewards individuals which maintain a close to constant number of magnets in the *on* state. Additionally we employ a penalty when the number of unique states in the trajectory is lower than the total length of the trajectory, e.g, the trajectory settles into looping or stationary behaviour. This penalty deters the EA from achieving a constant number of active magnets by utilising undesirable behaviour, such as frozen or oscillating dynamics. We use a penalty factor of $k = 5100$, equal to the number of nanomagnets in our simulated pinwheel ASI. A fitness score of 0 indicates perfect glider behaviour has been attained.

For mutation, the algorithm selects whether to mutate $H^+$, $H^-$ or the initial state with even probability. Field strengths are mutated using Gaussian mutation with standard deviation 0.025, with the resulting field bounded to the range [0.025, 0.05]. For mutation of the initial state list, one element in the list is selected at random and mutated using Gaussian mutation with standard deviation $\frac{2}{3}$, with the result bounded to the unit interval.

For the recombination of two individuals, point crossover is applied to the initial state list to produce two offspring. Each offspring copies the field strengths of a different parent.

A mutation rate of 0.2 and a crossover rate 0.3 is used. In each generation, 20% of the population is chosen for mutation and 30% is chosen for crossover. The mutants and offspring are then evaluated for fitness. Selection considers the offspring, mutants and unaltered population and retains the 100 individuals with best fitness.

## flatspin simulations

Simulations are carried out using the flatspin point-dipole ASI simulator[28]. In each simulation, the clock protocol *aAbB* was sequentially applied 20 times to a $50 \times 50$ pinwheel ASI consisting of 5100 nanomagnets. The switching astroid of the simulated nanomagnets correspond to that of stadium shaped nanomagnets of dimension $220\,nm \times 80\,nm \times 10\,nm$.

The flatspin simulations, used as part of the EA and in the subsequent field strength exploration, were run with the following parameters: $b = 0.4040$, $c = 1.0$, $\beta = 1.7331$, $\gamma = 3.5195$, $h_k = 0.1303$, neighbour distance = 10, lattice spacing = 1. The values of $b$, $c$, $\beta$, $\gamma$ and $h_k$ are taken from the astroid database provided with flatspin, for stadium magnets of size $220\,nm \times 80\,nm \times 10\,nm$, with a saturation magnetisation value of $860\,kA\,m^{-1}$. For the evolutionary run, a high $\alpha$ value of 0.0025 was used to encourage local interactions. In all other subsequent simulations and analysis, $\alpha = 0.001627$ was used as it was found to most closely agree with MuMax3 simulations of the experimental system.

Applying fields to an ASI often causes nucleation at the edges which grows inwards, due to the edge magnets having fewer close neighbours stabilising them. However, we wish the nucleation to be governed by the evolved initial state rather than the edges. To this end, we add two layers (one of each sublattice) of buffer magnets around the edge of the array which are made harder to switch by increasing $h_k$ by a factor 10.

For the snake shift register demonstrated in Supplementary Movies 8 & 9, four layers of buffer magnets were needed. A shift register of length $N$ will have a memory capacity $MC = N - 1$, as each input $u_t$ is present in the register for $N$ time steps (after which it is shifted out when reaching the end of the register). Using a snake length of two to encode each bit, the length of our snake shift register is $N = \frac{W-2}{2}$ where $W$ is the width of the ASI. We subtract two to account for buffer magnets at the rightmost edge. Correspondingly, the resulting $MC = W/2 - 2$ for the snake shift register.

When simulating with disorder, the modified switching threshold $h_k'$ for each magnet is sampled from the normal distribution $\mathcal{N}(h_k, d \cdot h_k)$, where the disorder variable $d$ scales the level of disorder. Each simulation was run 30 times with different random seeds to produce statistics on the effect of different instances of disorder. Specifically, we compare the snake's starting position to the final position it reaches while still resembling a snake, i.e., it consists of only two rows, with a head and a tail. As this definition allows for the snake to grow and shrink while moving, we measure the distance travelled by the head and tail of the snake separately, and take the minimum of these two values to be the true distance travelled.

## Sample fabrication

For experimental demonstration, we fabricated a $100 \times 100$ pinwheel ASI consisting of $220\,nm \times 80\,nm \times 10\,nm$ stadium-shaped Permalloy nanomagnets, with a lattice spacing of 247.5 nm. Two layers of buffer edge-magnets were realised using $220\,nm \times 70\,nm \times 10\,nm$ stadium-shaped nanomagnets, where the altered aspect ratio results in a higher switching field.

The pinwheel ASI with buffer edge-magnets was fabricated using an electron beam lithography lift-off process. A 1:2 CSAR 62:anisole electron resist mixture was spin coated onto a Si substrate at 3250 rpm, resulting in a 90 nm thick electron resist layer. We then soft-baked the sample at $150°\,C$ for 1 min, before exposing the desired pinwheel ASI using an Elionix ELS-G100 EBL system. Following exposure, the resist layer is developed in AR600-546 for 40 s. We then deposited a 10 nm layer of Permalloy ($Ni_{0.81}Fe_{0.19}$) using a K.J. Lesker E-beam evaporator. Finally, ultrasound-assisted lift-off was performed in AR600-71.

## Experimental demonstration

We experimentally demonstrate movement of the snake with the *aAbB* protocol by applying clock fields using an in-plane quadrupole vector magnet, where each magnetic field is applied for a couple of seconds. The magnetic state of the pinwheel ASI is imaged using magnetic force microscopy (MFM) after each applied field.

We first polarise the pinwheel ASI with a 60 mT field pulse along 180°. The snake state is initialised by writing a single $\approx 1\,\mu m$ line, placed between sublattice $L_a$ and $L_b$, with the MFM-tip in the presence of a 10 mT bias field along 0°. The 10 mT bias field is not strong enough to switch any nanomagnets on its own, but facilitates consistent and reliable magnetic writing with the MFM-tip. Writing was performed with the MFM-tip in contact with the sample, at a scan speed of $55\,\mu m\,s^{-1}$. We then apply a series of in-plane fields at the relevant clock field angles but with slightly varying field strengths to overcome some particularly hard to switch magnets and to optimise the field strengths used for clocking.

To verify the experimental robustness of the snakes, we initialised groups of 5-6 snakes at a time. The 5-6 snakes were initialised on every fifth row of the pinwheel ASI to avoid interaction between neighbouring snakes while maintaining the possibility of imaging several snakes simultaneously. All initial snakes were of the same orientation (with the nanomagnets in the top row of the snake belonging to sublattice $L_a$). The snakes were then clocked using the *aAbB* field protocol, until they stopped moving. The pinwheel ASI was then polarised and the same procedure was repeated, with either new snakes (shifted one row up or down) or new clock field strengths.

Both writing the initial state and imaging were done using commercial MFM-probes (NanoWorld POINTPROBE MFMR). MFM-imaging was done at remanence, with 55 nm to 60 nm lift height and $50\,\mu m\,s^{-1}$ to $55\,\mu m\,s^{-1}$ scan speed. All experiments were performed at room temperature.

## Data availability

The MFM data and MuMax3 input file used in this study have been deposited in the Zenodo database at https://doi.org/10.5281/zenodo.15055562.

## Code availability

Numerical simulations were performed using the open-source simulators flatspin (https://flatspin.gitlab.io/) and MuMax3 (https://mumax.github.io/).

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

## Acknowledgements
This work was funded in part by the Norwegian Research Council TEKNOKONVERGENS project SPrINTER (Grant no. 331821), and in part by the EU FET-Open RIA project SpinENGINE (Grant no. 861618). Simulations were executed on the NTNU EPIC compute cluster[39]. The Research Council of Norway is acknowledged for the support to the Norwegian Micro- and Nano-Fabrication Facility, NorFab, project number 295864.

## Author contributions
A.P. made the initial discovery and designed the evolutionary experiment, with contributions from G.T. A.P. and J.H.J. designed the simulation study and performed the analysis. I.B. fabricated the samples and, together with A.S., did the experimental work. E.F. and G.T. provided feedback and suggestions. A.P., J.H.J., I.B. and A.S. wrote the manuscript with input from all authors.

## Funding
 Olavs Hospital - Trondheim University Hospital).

## Competing interests
The authors declare no competing interests.
