## [Transparent Peer Review file · Nature Communications]

Controllable Gliders in a Nanomagnetic Metamaterial

Corresponding Author: Dr Arthur Penty

Version 0:

Reviewer comments:

Reviewer #1

(Remarks to the Author)

This work reports the experimental observation of a special type of collective motion of magnetic textures in artificial spin ice (ASI), named as "snake". Under certain field protocol, the authors show that the snake can translate across the ASI lattice while maintaining its shape, with its direction determined by its initial orientation. The behavior observed is similar to that of a glider, which is a simple structure of cellular automaton, promising for information transmission, storage and computation. The authors validate their experimental discovery via both flatspin and micromagnetic (MuMax3) simulations. The topic of this work is interesting and the idea of realizing a physical glider of a Cellular Automaton is novel. However, I don't recommend acceptance of the paper for the following reasons:

1. Although the snake observed in this work and the glider in cellular automata are similar in their self-preserving movement characteristics, their underlying mechanisms are fundamentally different. In cellular automata, the glider's self-preserving nature emerges from precise local rules (such as survival with two to three neighbors, death under underpopulation or overpopulation, and reproduction), which enable movement not just horizontally or vertically, but also diagonally, thereby allowing cellular automata to simulate complex systems and perform computational tasks. In contrast, the snake in this ASI system seems more like a collective magnetic behavior, where the local rules of individual units are not clearly defined. There are statements like "The smallest functional snake is of length two... There is no upper-bound on the length of the snake", indicating that the snake appears more as an emergent phenomenon of multiple spin ice islands acting as a magnetic collective, rather than a strictly defined, rule-based moving structure. Another indication is that although a 2D ASI system is investigated, the snake can only move horizontally, not diagonally, suggesting their motion is constrained by orientation-specific, collective rules rather than purely local interactions.
2. The authors did discuss the microscopic mechanism of the snake, such as the crucial role of neighbor interactions (both nearest and next-nearest) and the specific roles of dipolar interactions in both stabilizing and destabilizing different parts of the snake. The discussion of neighbors' influence in Fig.4 is hand-waving. Although this work provides a qualitative understanding through color-coding and descriptive terms like "strongly stabilizing" or "weakly destabilizing", it fails to provide quantitative analysis such as numerical values of dipolar field strengths, phase diagram of interaction energies and specific threshold values for switching events etc. In contrast to the qualitative neighbor analysis presented in the paper, what would be expected for a rigorous treatment is a quantitative set of deterministic rules similar to Conway's Game of Life cellular automaton, where precise numerical criteria (e.g., "a live cell with exactly two or three live neighbors survives") definitively determine cell state transitions. Such quantitative rules for neighbor-dependent switching are notably absent in the current magnetic neighbor analysis. Note that here "neighbor" means neighbor cells of a cellular automata, not neighbor islands of artificial spin ice.
3. There is no explicit definition of what constitutes a "cell" in this work. The fundamental unit described is the individual nanomagnet (magnetic island), which behaves as a binary artificial spin. While the work draws parallels to cellular automata, this appears to be a conceptual comparison about system characteristics (many simple elements with local interactions) rather than a direct mapping of CA cells to physical structures.
4. In my opinion, this work is closely related to monopole dynamics in ASI system, but a comprehensive review is lacking. Also, the discussion on neuromorphic computing is too narrow, there have been many realizations of reservoir computing based on similar field-driven protocols, which have not been properly cited.

5. The computational capabilities of the proposed magnetic snake-like structure remain unclear. The demonstrated behavior appears fundamentally similar to domain-wall racetrack memory, primarily offering information storage and transmission rather than computation. The distinction between simple information propagation and actual computation requires further clarification.

In summary, the novelty of cellular automaton lies in achieving complex global dynamics through simple and well-defined local rules. However, the manuscript does not show convincingly that the observed snake structure shares the same underlying mechanisms as gliders in cellular automata. Therefore, the phenomenon appears to be an interesting collective motion in ASI, making it more appropriate for publication in a specialized journal.

Reviewer #2

(Remarks to the Author)
See attached file.

Reviewer #3

(Remarks to the Author)
I have attached a referee report.

Reviewer #4

(Remarks to the Author)
I co-reviewed this manuscript with one of the reviewers who provided the listed reports. This is part of the Nature Communications initiative to facilitate training in peer review and to provide appropriate recognition for Early Career Researchers who co-review manuscripts.

Reviewer #5

(Remarks to the Author)
This is an exciting paper that shows, for the first time, the ability to propagate magnetic textures through 2D lattices of interacting nanomagnets (ASIs). I agree with the authors on the significance of their work; the ASI system has always had a superficial resemblance of the MCA that produce elegant emergent behaviour in e.g. the game of life, but previously no researchers had shown that this could be genuinely realised. The ability to produce propagating gliders is a substantial step forward in that direction. Furthermore, I do think the ability to propagate information could have significant impacts on attempts to use ASIs as substrates for reservoir computing. Overall, this seems a original and impactful study suitable for publication in nature comms.

I do feel the papers results could be expanded a little. For example, the authors discuss "multiple interesting structures with glider like properties" it would be good to explore these in paper, or at least in the supplementary material. They mention how snakes of different lengths can be propagated, and it would be good for that to be evidenced. The experimental results are rather sparse, and the figure in the paper shows an even smaller range of motion than the video in the supplementary materials. Is there any way this could be expanded a little to show the robustness of the approach? Finally, they comment that the snake is like a shift register, but operating in a 2D lattice. It is, but it only moves in one axis - can they perhaps speculate on how full 2D control might be gained?

I also have a few minor points that it would be good to address:

(1) I found the sentence "whose nanomagnets are rotated $+45^\circ$ and -45° , respectively. In this work, we refer to the sublattice and magnetisation of the nanomagnets by their colour (blue or orange for La and green or pink for Lb), as indicated by the coloured inset in Fig. 1a" a bit confusing. Firstly isn't it a rotating of -45° or 45° degrees? Secondly, I found the explanation of the colour scheme a little confusing. Can they expand this a bit for clarity?

(2) On page 4 "EA" should be defined.

Overall, this is good work and a significant step for the control of ASIs towards interesting emergent behaviours, and I'd like to see it published in Nature comms, especially if its results could be expanded a little.

Version 1:

Reviewer comments:

Reviewer #1

(Remarks to the Author)

I would like to thank the authors for their thorough response to my previous comments and the substantial revisions made to the manuscript. The additions of the new "Robustness" and "Toward computation" sections, along with the improved organization and clarification in the introduction, have undoubtedly enhanced the quality and readability of the manuscript.

However, after reviewing the authors' response and the revised manuscript, I find myself more convinced of my initial assessment. The authors themselves acknowledge key points that support my original concerns: The authors explicitly state: "It is indeed true that the underlying mechanism for the game of life (GoL) and the snake are different." They further clarify that "the goal of this work is not to perfectly emulate a cellular automata (CA) in ASI." This admission confirms my original position that the work presents a "glider-like" behavior in ASI rather than a true implementation of CA gliders. This difference is important when considering the actual contribution of this work. The title "Glider in Artificial Spin Ice" and numerous references to cellular automata throughout the manuscript remain potentially misleading. A more accurate title would be something like "Snake with Glider-like Behavior in Artificial Spin Ice," which would better reflect the actual nature of the discovery.

I acknowledge that the authors have discovered a novel and interesting collective motion pattern in ASI that exhibits some properties similar to gliders. This finding is different from previously reported phenomena such as monopole dynamics and represents a new contribution to the field. However, I maintain that the significance of this work is substantially lower than what would be expected for a true implementation of cellular automata gliders in ASI. The newly added sections on robustness and computational capabilities, while informative, do not fundamentally alter this assessment.

Reviewer #2

(Remarks to the Author)

I thank the authors for their carefully considered responses and updates to the manuscript. They have adequately addressed my concerns and those of other reviewers. The additional sections better place this work in the context of computing.

I just have one small addition. At the bottom of page 6 in the "Experimental demonstration" section the authors write:

"The initial snake state is written onto a polarised ASI using a magnetic force microscopy (MFM) tip."

There are two papers that describe this kind of tip writing and should be cited in the text. The authors should also state whether this was done with or without an applied magnetic field. The papers are:

Yong-Lei Wang et al. ,Rewritable artificial magnetic charge ice.Science352,962-966(2016).DOI:10.1126/science.aad8037

Gartside, J.C., Arroo, D.M., Burn, D.M. et al. Realization of ground state in artificial kagome spin ice via topological defect-driven magnetic writing. Nature Nanotech 13, 53–58 (2018). <https://doi.org/10.1038/s41565-017-0002-1>

After including these references I recommend publication.

Reviewer #3

(Remarks to the Author)

I have read with interest the revised version of the manuscript by Penty et al., and I commend the authors for the significant improvements made. The inclusion of the "Robustness" and "Towards Computing" sections enhances both the scientific depth and relevance of the work. The revised manuscript now more clearly articulates the rationale for investigating the "snake glider" as a platform for information transmission, storage, and transformation in artificial spin ice (ASI).

One of the central contributions of this work is the demonstration that controlled, glider-like motion is possible in ASI under astroid clocking, an achievement that may hold substantial implications for information processing in nanomagnetic systems. I find the development of a global clocking protocol that enables deterministic movement of a domain-like excitation ("the snake") particularly compelling—especially given its robustness to certain levels of disorder and its ability to self-correct after moderate defects.

That said, a number of conceptual and technical points merit further clarification or expansion.

Major Points:

While the paper discusses the computing potential of the snake glider and its qualitative advantages over earlier ASI-based approaches, I still find that it falls short in substantiating these claims through quantitative benchmarks. The "Towards Computing" section does a better job than the previous version in highlighting how the snake could serve as a memory or logic element. However, if the authors aim to assert that the snake glider can function as an effective substrate for reservoir computing (RC), it would strengthen the manuscript considerably to include results on an established RC task—such as predicting the Mackey–Glass time series at various horizons ($t+10$, $t+20$, etc.) or computing memory capacity curves as in Jaeger (2001).

As it stands, the paper gestures toward the potential for RC but lacks a direct test of this hypothesis. This is particularly important given the relatively bold claims made about integrating memory, transmission, and logic in one substrate.

I appreciate the authors' inclusion of references to Gartside et al. (2022) and Stenning et al. (2024), who showed that the trade-off between memory and nonlinearity in ASI can be mitigated via multilayer or multi-modal architectures. However, I still feel this prior work should be more explicitly contrasted with the current approach. For example: Is the snake glider operating in a different part of the design space? Does it offer advantages in energy efficiency, reconfigurability, or scalability that are not achievable through spin-wave-based RC?

Perhaps the most intriguing aspect of the manuscript is the ability to control the glider's motion deterministically through clock fields, which elevates this from a curiosity to a potential building block. This level of control is a nontrivial step forward and deserves further emphasis. It would be beneficial to explain whether this level of control could be extended to more complex logic operations or routing schemes in a 2D ASI network.

In traditional RC or neuromorphic paradigms, learning is often implemented through feedback and adaptation of internal weights. In contrast, the snake glider system relies on structure-driven dynamics governed by externally applied fields. It would be helpful if the authors could clarify whether, and how, feedback-driven learning might be realized in this framework—for instance, via adaptive modulation of clock field strength, snake initialization, or environmental inputs. This would help bridge the gap between biological inspiration and functional implementation.

Minor Comments:

The use of the term "glider" is appropriate and well-justified through analogy with cellular automata, but I recommend being careful not to overstate the equivalence. The emergent dynamics are clearly more constrained than in Game of Life, where gliders can interact in richer ways. That said, the glider analogy remains helpful and illustrative.

Figure 8, showing collision dynamics, is one of the most compelling demonstrations in the paper. If possible, it would be valuable to elaborate on whether these transformations can be harnessed in a programmable fashion, or if they remain stochastic due to disorder.

The revised introduction is clearer, though still somewhat long. The discussion of prior limitations in ASI computing should come earlier, to better motivate the problem the authors address.

Recommendation:

While I believe this work represents a meaningful and creative step in ASI-based information processing, I still have some reservations about its readiness for Nature Communications. In particular, I would expect either (1) a demonstration of a concrete computational benchmark using the snake glider, or (2) more extensive validation of its scalability and applicability to real-world neuromorphic computing tasks.

That said, this is clearly a high-quality contribution and one that would be well-suited for a specialized or emerging journal in the field, such as Communications Physics, Scientific Reports, or Advanced Intelligent Systems. Should the authors include a proof-of-principle RC task, I would be inclined to reconsider the suitability for Nature Communications.

Reviewer #4

(Remarks to the Author)

Reviewer #5

(Remarks to the Author)

The authors have made substantial efforts to improve their manuscript and it is improved as a result. I still find the experimental results slightly disappointing - it is clear that disorder in the array is greatly inhibiting the ability to reliably transmit gliders through the array. However, the additional measurements and simulations of the effects of disorder provided clear explanations and indications of the levels of disorder required to produce more reliable motion.

More generally the revisions have made the manuscript more thoughtful on the implications of their discoveries, and I found it interesting to read. I remain convinced it is a step forward for the field with interesting implications for computing research.

Overall, I think this now merits publication. It would be great if more reliable motion could be demonstrated, but I wouldn't want to inhibit this being released to the community by insisting on this.

Version 2:

Reviewer comments:

Reviewer #1

(Remarks to the Author)

I would like to thank the authors for their comprehensive response to my previous comments and the substantial revisions made to the revised manuscript. The authors have addressed my concerns regarding the definition of "glider" by making important clarifications and adding appropriate citations. Specifically, they have clarified their use of the term "glider" in accordance with Adamatzky (2010), stating that they are not implementing Conway's Game of Life in ASI but rather

searching for configurations where glider-like behavior can be found. The title change to "Controllable Gliders in a Nanomagnetic Metamaterial" and the added literature citations make the contribution more appropriately positioned and no longer misleading.

While I maintain some reservations about the potential for using gliders in truly meaningful computing tasks beyond the demonstrated examples, I acknowledge that the authors have shown compelling evidence of controllability and nonlinearity in these moving magnetic structures. The demonstrated shift register functionality and various collider behaviors indicate that this type of motion in magnetic structures does possess computational potential and manipulability. As the authors correctly point out, such foundational demonstrations are important stepping stones, and I do not insist that they demonstrate more extensive computational capabilities at this stage. Therefore, I recommend acceptance of this manuscript.

Reviewer #3

(Remarks to the Author)

The authors have satisfactorily answered all my queries. I do now think that this paper is eligible to be published in Nature Communication.

Reviewer #4

(Remarks to the Author)

Response to Reviewers

We would like to thank all reviewers for their valuable time and constructive feedback on our manuscript. Their insightful comments have enabled us to make significant improvements to the paper.

As a response to comments and questions from several reviewers we have included two new subsections in the Result section. The added "Robustness" subsection address robustness at a conceptual level of how to define quantifiable robustness for snake-like behaviour and simulation results for disorder's influence. Further, we have included experimental results to demonstrate robustness in the snake glider. To improve on principles for the computational aspects and properties for snake gliders we have added the subsection "Toward computation". In this new subsection the computational properties information transmission, storage and transformation are elaborated and put in the context of snake glider behaviour. This subsection now include simulations demonstrating information transformation and interaction between snake gliders.

Aside from these additions, the text has been reworded and reorganised in some parts as to clarify any potential misunderstandings for readers, highlighted by the reviewers. Furthermore the introduction has been refocussed to better present and contextualise the potential impacts of our results.

Feedback from Reviewer #1:

This work reports the experimental observation of a special type of collective motion of magnetic textures in artificial spin ice (ASI), named as "snake". Under certain field protocol, the authors show that the snake can translate across the ASI lattice while maintaining its shape, with its direction determined by its initial orientation. The behavior observed is similar to that of a glider, which is a simple structure of cellular automaton, promising for information transmission, storage and computation. The authors validate their experimental discovery via both flatspin and micromagnetic (MuMax3) simulations. The topic of this work is interesting and the idea of realizing a physical glider of a Cellular Automaton is novel. However, I don't recommend acceptance of the paper for the following reasons:

We thank the reviewer for noting the novelty of our work, and have attempted to rectify the concerns highlighted in their review.

Although the snake observed in this work and the glider in cellular automata are similar in their self-preserving movement characteristics, their underlying mechanisms are fundamentally different. In cellular automata, the glider's self-preserving nature emerges from precise local rules (such as survival with two to three neighbors, death under underpopulation or overpopulation, and reproduction), which enable movement not just horizontally or vertically, but also diagonally, thereby allowing cellular automata to simulate complex systems and perform computational tasks. In contrast, the snake in this ASI system seems more like a collective magnetic behavior, where the local rules of individual units are not clearly defined.

It is indeed true that the underlying mechanism for the game of life (GoL) and the snake are different. The snake's underlying mechanisms are dipolar interactions and reversal of nanomagnets, while GoL is an abstract model and thus has no underlying mechanism beyond its abstractly defined rule set. The common principles of CAs, including GoL, exploited in our approach are the principles of vast number of simple elements (CA-cells, nanomagnets), local interactions, parallel operation and discrete time dynamics.

The glider in GoL is defined by an initial structure that can move in four directions (the four diagonals) depending on the rotation of the initial structure. Similarly, the snake can move left or right depending on the rotation of the initial snake. GoL's underlying rule table consisting of 2^9 10-bit binary rules, that can abstracted to the succinct textual description, also define the glider in GoL. The snake's underlying rules can similarly be abstracted from the underlying dipolar interactions and reversal of nanomagnets to a set of behaviours far less then the underlying state space of the nanomagnets. We have added a new figure (Fig. 3) and outlined a schematic description of how the snake's structure changes from one state to the next state, i.e., $t_n \rightarrow t_{n+1}$. We would consider the glider an emergent property in GoL, it is not specified in the rules yet emerges as a consequence. Similarly, the snake's glider behaviour emerges as a consequence of the underlying mechanism.

Furthermore, the goal of this work is not to perfectly emulate a cellular automata (CA) in ASI, but to reproduce the glider-like properties due to the many benefits they bring with them. We felt this was not

clear in our submission, so we have made changes to the introduction to highlight our goal.

There are statements like “The smallest functional snake is of length two... There is no upper-bound on the length of the snake”, indicating that the snake appears more as an emergent phenomenon of multiple spin ice islands acting as a magnetic collective, rather than a strictly defined, rule-based moving structure. Another indication is that although a 2D ASI system is investigated, the snake can only move horizontally, not diagonally, suggesting their motion is constrained by orientation-specific, collective rules rather than purely local interactions.

The reason the snake can be at any length is because the head and tail function independently. We have added a table (Fig. 3) showing the rules followed by the head and tail. In Fig. 5 we show that considering up to the 2nd nearest neighbour contributions accounts for the snakes behaviour. Additionally, we have seen in flatspin simulation that we can reduce the neighbour distance, such that only 1st and 2nd neighbours are considered, and we still see the snake behaviour.

2. The authors did discuss the microscopic mechanism of the snake, such as the crucial role of neighbor interactions (both nearest and next-nearest) and the specific roles of dipolar interactions in both stabilizing and destabilizing different parts of the snake. The discussion of neighbors’ influence in Fig.4 is hand-waving. Although this work provides a qualitative understanding through color-coding and descriptive terms like “strongly stabilizing” or “weakly destabilizing”, it fails to provide quantitative analysis such as numerical values of dipolar field strengths, phase diagram of interaction energies and specific threshold values for switching events etc.

We thank the reviewer for their suggestion to provide a more quantitative analysis. In Fig. 5 (previously Fig. 4) we have added a colour bar to quantify the stabilisation, and give the corresponding dipolar field values in the text. Furthermore, the 0% disorder plot in the new Fig. 6, provides a quantitative understanding of the range of field values in which the snake functions.

In contrast to the qualitative neighbor analysis presented in the paper, what would be expected for a rigorous treatment is a quantitative set of deterministic rules similar to Conway’s Game of Life cellular automaton, where precise numerical criteria (e.g., “a live cell with exactly two or three live neighbors survives”) definitively determine cell state transitions. Such quantitative rules for neighbor-dependent switching are notably absent in the current magnetic neighbor analysis. Note that here “neighbor” means neighbor cells of a cellular automata, not neighbor islands of artificial spin ice.

The newly added Fig. 3 shows the deterministic behaviour of the head and tail, provided the field strength are within the ranges given in Fig. 6. While it is true that the GoL rule set can be succinctly described in a sentence this is not a requirement for CA. In fact an explicit rule set for GoL would consist of 2^9 rules.

3. There is no explicit definition of what constitutes a “cell” in this work. The fundamental unit described is the individual nanomagnet (magnetic island), which behaves as a binary artificial spin. While the work draws parallels to cellular automata, this appears to be a conceptual comparison about system characteristics (many simple elements with local interactions) rather than a direct mapping of CA cells to physical structures.

We think it make sense to consider one magnet as a single 2-state CA cell, however this choice is arbitrary. One could, just as easily, consider each pinwheel unit a cell, which would map to a 2^4 -state CA. As we hope we have clarified with this new submission, our goal is not to create CA in ASI, and as such, discussion of cell size is not directly relevant.

4. In my opinion, this work is closely related to monopole dynamics in ASI system, but a comprehensive review is lacking.

We agree that there are apparent similarities with monopole dynamics, as such we have sought to highlight the differences between our work and that of monopole dynamics. We have included a sentence in the introduction stating why we are looking for something other than monopole dynamics. In the analysis

section we give a brief explanation of how the snake cannot be understood purely through monopole dynamics, and a more extensive explanation in the supplementary materials.

Also, the discussion on neuromorphic computing is too narrow, there have been many realizations of reservoir computing based on similar field-driven protocols, which have not been properly cited.

We have attempted to expand our neuromorphic discussion while also balancing other reviewers' requests for a more streamlined introduction. We have added some additional citations of examples of ASI reservoir computers.

5. The computational capabilities of the proposed magnetic snake-like structure remain unclear. The demonstrated behavior appears fundamentally similar to domain-wall racetrack memory, primarily offering information storage and transmission rather than computation. The distinction between simple information propagation and actual computation requires further clarification.

We have added a new section, "Towards Computing", which demonstrates information transmission, transformation and storage through use of the snake and snake collision. The reviewer is correct that, without these interactions, the glider would just be performing the same as a racetrack memory. But with the snakes being able to interact with structure above and below them, they are able to make use of the 2D array and perform information transformation.

In summary, the novelty of cellular automaton lies in achieving complex global dynamics through simple and well-defined local rules. However, the manuscript does not show convincingly that the observed snake structure shares the same underlying mechanisms as gliders in cellular automata. Therefore, the phenomenon appears to be an interesting collective motion in ASI, making it more appropriate for publication in a specialized journal.

We thank the reviewer for their constructive feedback. We have made improvements and added material to address the comments in the review. We have re-written the introduction to clarify that our goal is not to make a CA in ASI but to find a structure in ASI which matches phenomenologically to the behaviour and computational implications of a glider in CA. We have also added more motivation as to why the glider phenomenon, at the computational and "emergent" magnetic level, in ASI would be of interest to a wider audience.

Feedback from Reviewer #2:

The authors have demonstrated through simulation and experiment how to create cellular automata (CA) in a pinwheel artificial spin ice (ASI). The authors used an evolutionary algorithm to discover an equivalent CA to the "glider" in Conway's Game of Life which they call a "snake". They demonstrate the linear movement of a "snake" via a simple "astroid clocking" field protocol. They explain how the dipolar field of nearest neighbours stabilise and destabilise the ends of the "snake" to allow it to transmit through the lattice in a stable way. The authors argue that this will be of use to ASI based computing which currently lacks short term memory. I think this is fantastic work and recommend publication after some minor corrections and a suggested expansion of the discussion.

We thank the reviewer for their enthusiasm and have strived to carry out the corrections they suggest.

I think the use of time-evolution (dynamics) could be misinterpreted as magnon or spin-waves by some. I suggest (reversal-dynamics) or something similar.

To avoid any confusion we have replaced the term with "ASI trajectory" and "movement".

After carefully following the authors analysis of the nearest neighbour's dipolar field effect on the snake I think the effective snake looks something like this:

Should the snake be considered as only the reversed magnets (red box) or the surrounding nearest neighbours also (black box)? I think an indication of an effective boundary for the

snake will be useful for the reader.

We have added an inset to the top of Fig. 2 to show clearly the extent of the snake glider and clear up any confusion.

It is a shame that the “pointy” end and the tail is with the “forked” end. Intuitively I think of a snake’s tail as being pointy and having a forked tongue for the head. The authors should either pick different nomenclature or carefully describe what they mean by pointy and forked to avoid reader confusion. Perhaps using heads and tails of an arrow would be better, but this is just a suggestion.

We have added the arrow analogy as suggested, and with the new inset in Fig. 2 we think this will be clear.

Are the authors able to elaborate on how they would nucleate a functional snake? The authors point out that the smallest functional snake has length two where there are two magnets from each sublattice. Is it possible to create the snake structure from the above left? Here the tail and head are effectively the same. But if it is possible to do this you should then be able to change the direction of a snake shrinking to the left structure and then rotating the clock fields 90 degrees. Perhaps a different clock sequence is required to transform between the two states below. I think if the authors can demonstrate this functionality, it would vastly increase the impact of the paper since now you are making use of the 2D structure more fully.

The authors could cite some papers on how initialisation of structures could be achieved in practice. For example:

Nature Nanotechnology volume 13, pages 53–58 (2018) Cell Reports Physical Science, Volume 4, Issue 3, 15 March 2023, 101291

Communications Physics volume 3, Article number: 219 (2020)

While it is possible to shrink the snake down to the structure shown, vertical movement is not so trivial as simply rotating the fields. In the case of the horizontal moving snake, the leftwards polarised snake is moving with or against a rightwards polarised background domain. For vertical movement, the snake would be moving perpendicular to the background domain. While it may be possible with some field protocol, it would rely on a different mechanism.

We have added a sentence in the Discussion which highlights how snake gliders might be nucleated in a practical device, based on the references provided.

The authors mention that the nearest neighbour dipolar fields push the switching fields outside the switching asteroid. Is there a way to indicate the effect on an asteroid plot for the cases shown in fig 4. This could be included as supplementary information.

We have added a figure in the supplementary showing this.

The following is probably beyond the scope of the current study but I think it would be useful to include some answers to these as possible future directions. In the game of life, the rules are set and the complex behaviour emerges from the initialised state. The authors use an evolutionary algorithm that varies both initial state and the clock fields. For a fixed clock field do the authors find other cellular automaton? Ideally you would want the same clock algorithm to apply to get the complex computations from the initialised state alone. If the clock fields used here can apply to other structures this would be very exciting.

The authors also neglect structures with frozen or oscillatory behaviour but these can be instrumental in generating snakes. For example, the Gosper glider gun in GoL consists of static and oscillatory elements to produce a steady stream of gliders. Can the EA’s fitness function be modified to reward an oscillating structure that also generates snakes indefinitely?

We do indeed see different structures capable of “surviving” under the same clock fields. For example in the new Fig. 7 we see two snakes collide to create a still life, showing that this still life structure and the snake can inhabit the same ASI “CA”. We give further examples of some other gliders in the supplementary. Regarding the Gosper gun, this is absolutely something we are interested in and pursuing,

however this will be a significant undertaking and is outside the scope of this paper.

Feedback from Reviewer #3:

I read with great interest the manuscript by Penty et al. I was intrigued by the title at first. The manuscript is undoubtedly intriguing as a whole, and contains several valuable contributions, particularly in exploring the role of "snake gliders" in pinwheel, a type of artificial spin ice lattice. The work builds on prior knowledge mostly developed previously by a subset of these authors, and by creatively applying the concept of gliders to the ASI system and provides a comprehensive experimental and theoretical analysis. However, as much as the manuscript is well-constructed, it leaves the reader wanting more—there is a certain lack of impact or a "punch" that would elevate this work from a solid contribution to a standout paper. Below, I elaborate on key areas where improvements could be made to enhance the clarity, accessibility, and scientific significance of the manuscript. I also attach my comments directly to the text below. My report below should be read within the context of those comments.

We thank the reviewer for their interest and extensive review. We hope that with the updated introduction and the new sections investigating the robustness and computational potential of the snake, the manuscript now provides the desired "punch".

First, while the concept of the snake glider is well-motivated by parallels to cellular automata and the Game of Life, the broader significance of these endings for computation is not fully developed. The idea of leveraging ASI for computation has been explored before, including these authors, but this paper seems to leave the potential applications of the snake glider in neuromorphic or unconventional computing somewhat underexplored.

We have added a new paragraph to the introduction speculating on how the snake glider could be of use beyond neuromorphic computing. Also, we have added a new section, "Towards Computing", which further considers how the snake can be used for computing.

While the authors suggest that the glider can transmit and store information, there is a missed opportunity to more explicitly position this as a breakthrough in integrating memory and information transfer in ASI systems. For instance, can this glider outperform other proposed mechanisms for memory in ASI?

In the new Towards Computing section, we give a brief explanation of how the snake outperforms current ASI reservoirs in terms of memory capacity, and provide video demonstrations in Supplementary Movies 8 & 9. Compared to pure memory implementations, the snake provides both memory and computation, which could form the basis for in-memory computing.

Is it scalable or robust enough to serve as a basis for real-world implementations? These points need clearer articulation. I see value in this contribution, but at the same time it leaves me wondering how to use it.

We have added a new Robustness section, which investigates and demonstrate robustness in simulation and experiment.

Second, the authors describe the evolutionary algorithm (EA) used to discover the snake glider, but the explanation lacks sufficient detail to be convincing to the reader. The mechanism of the EA, the fitness function, and the parameter optimization process are only briefly described and would benefit from greater transparency instead of being buried in the Methods. For example, what were the key constraints or challenges in finding a stable glider, and how were these overcome? Including a more thorough discussion of the EA's success or failure in identifying other gliders would provide additional depth and allow the reader to understand whether this method could be generalized for other magnetic systems.

We have added two paragraphs to the Results section explaining the EA and the considerations that went into constructing the fitness function.

The manuscript would also benefit from a more direct comparison with prior work in the field, at least at the level of context. While there are scattered references to studies on domain growth and magnetic monopoles, these feel somewhat disconnected from the main narrative of computing. A stronger engagement with foundational work in ASI and neuromorphic computing would help clarify what differentiates this contribution. I did add some references, feel free to pick from them.

We thank the reviewer for their suggested references. We have used them where applicable.

A related point is that the introduction, while thorough in describing the basic principles of ASI and cellular automata, feels somewhat long-winded and does not lead to a compelling research question. The problem of information storage and transmission in ASI is well-known, but the authors miss the chance to highlight the gap their work is addressing. The introduction could be streamlined and restructured to foreground the novelty of the snake glider concept and its broader implications.

We have streamlined the introduction and focussed on the benefits of a glider to computation. We have also added a new paragraph in the introduction describing the broader implications of the snake.

Furthermore, while the experimental validation of the snake glider is impressive, it remains unclear whether the demonstrated robustness is sufficient for practical applications. The discussion notes that fabrication defects and coercive field variations can break down the glider's motion, but does not provide a sense of how frequently this occurs or how it could be mitigated.

We have added a new Robustness section which investigates this point.

Similarly, while the authors show that the snake is capable of bi-directional movement, the potential for interacting snakes or more complex dynamics is not explored. This feels like a missed opportunity to extend the work beyond the relatively simple demonstrations shown in the manuscript.

We have added a new Towards Computing section, which demonstrates these interactions.

Lastly, while the figures and visualizations are generally clear and well-presented, there is a lack of intuitive explanation for some of the observed phenomena. For example, the dipolar interactions that govern the snake's movement are described in detail, but the connection between these interactions and the macroscopic glider motion is not made intuitive. A more conceptual or diagrammatic explanation of why the snake moves and why its head and tail behave differently would be a welcome addition.

We have added a new figure (Fig. 3) which gives a diagrammatic explanation of how the head and tail move under the different clocking fields.

In summary, while this work provides an innovative demonstration of gliders in ASI and is technically sound, it does not quite achieve its full potential. The manuscript is missing the critical impact or conceptual breakthrough that would make it truly memorable. By strengthening the connection to prior work, expanding the discussion of potential applications, and better contextualizing the importance of the findings, the authors could elevate this work to a higher level. For now, the paper reads as an interesting but somewhat niche contribution, rather than a transformative piece of research. I hope the authors will take these suggestions as constructive feedback to refine their message and broaden the appeal of their work.

There is a history of works similar to this one being published in Nature Communication. However, at this stage I honestly think that this is subpar compared to those - although I liked this paper and its innovative way of using/bring CA ideas into this field. This is definitely a good attempt/achievement towards this, I am just not sure if it is quite there to deserve a Nature Communication publication. I could see this published in Comm in Phys after a minor revision.

We thank the reviewer for their many helpful suggestions, and hope the new additions convince them of

the manuscript's potential impact.

Feedback from Reviewer #5:

This is an exciting paper that shows, for the first time, the ability to propagate magnetic textures through 2D lattices of interacting nanomagnets (ASIs). I agree with the authors on the significance of their work; the ASI system has always had a superficial resemblance of the MCA that produce elegant emergent behaviour in e.g. the game of life, but previously no researchers had shown that this could be genuinely realised. The ability to produce propagating gliders is a substantial step forward in that direction. Furthermore, I do think the ability to propagate information could have significant impacts on attempts to use ASIs as substrates for reservoir computing. Overall, this seems a original and impactful study suitable for publication in nature comms.

We thank the reviewer for their recommendation and we are grateful for the positive comments on the work.

I do feel the papers results could be expanded a little. For example, the authors discuss "multiple interesting structures with glider like properties" it would be good to explore these in paper, or at least in the supplementary material.

We have added three other gliders to the supplementary.

They mention how snakes of different lengths can be propagated, and it would be good for that to be evidenced.

We use the 5-snake in initial demonstration of glider movement, and the 3-snake in the new Towards Computing section. Experimentally, due to the initialization process, the snake length varies slightly. Additionally, we explicitly argue that the head and tail move independently, highlighted by Fig. 3.

The experimental results are rather sparse, and the figure in the paper shows an even smaller range of motion than the video in the supplementary materials. Is there any way this could be expanded a little to show the robustness of the approach?

We have added a Robustness section investigating this and providing more experimental results.

Finally, they comment that the snake is like a shift register, but operating in a 2D lattice. It is, but it only moves in one axis - can they perhaps speculate on how full 2D control might be gained?

In the Towards Computing section we show that, while the snake only moves one dimensionally, it can interact with structures above and below it. As such, despite being confined to one row, it can still make use of the 2D array. One could even envisage interactions whereby a snake moves up or down a row, or spawns a new snake on a different row. However this remains future work.

I also have a few minor points that it would be good to address:

*(1) I found the sentence "whose nanomagnets are rotated $+45^\circ$ and -45° , respectively. In this work, we refer to the sublattice and magnetisation of the nanomagnets by their colour (blue or orange for La and green or pink for Lb), as indicated by the coloured inset in Fig. 1a" a bit confusing. Firstly isn't it a rotating of -45° *or* 45° degrees?*

Secondly, I found the explanation of the colour scheme a little confusing. Can they expand this a bit for clarity?

We have changed the "and" to "or", and we have tried to clarify the colour scheme in text.

(2) On page 4 "EA" should be defined.

We have added this.

Overall, this is good work and a significant step for the control of ASIs towards interesting

emergent behaviours, and I'd like to see it published in Nature comms, especially if its results could be expanded a little.

We thank the reviewer for their recommendation, and the valuable feedback provided.

Response to Reviewers

We again thank all reviewers for their valuable time and constructive feedback on the revised version of our manuscript. Based on the feedback, we have updated the manuscript to expand and clarify several key areas. A summary of the changes are listed below.

- Changed the title to “Controllable Gliders in a Nanomagnetic Metamaterial”, to conform to journal guidelines.
- Added a reference to Dally et al. (2020) regarding the energy required for memory operations (Introduction).
- Added a reference to Adamatzky (2010) regarding the definition of “glider” in a CA (Introduction).
- Added two references (Wang et al., 2016; Gartside et al., 2018) regarding writing ASI state with an MFM tip (Experimental demonstration).
- Added memory capacity benchmark of a snake-based shift register (Towards Computing, Methods).
- Corrected a reference to Stenning et al. (2024) regarding the MC of width-modified pin-wheel (Towards Computing).
- Added a sentence emphasizing the deterministic property of the snake glider (Discussion).
- Added a sentence highlighting the scalability of memory capacity with snake gliders (Discussion).
- Added a sentence about the potential use case of the snake in a reconfigurable magnetic device (Discussion).

Please see our response to the feedback below, as well as the attached diff.pdf for a view of all changes.

Feedback from Reviewer #1:

I would like to thank the authors for their thorough response to my previous comments and the substantial revisions made to the manuscript. The additions of the new “Robustness” and “Toward computation” sections, along with the improved organization and clarification in the introduction, have undoubtedly enhanced the quality and readability of the manuscript.

We thank the reviewer for the feedback to improve the manuscript.

However, after reviewing the authors’ response and the revised manuscript, I find myself more convinced of my initial assessment. The authors themselves acknowledge key points that support my original concerns: The authors explicitly state: “It is indeed true that the underlying mechanism for the game of life (GoL) and the snake are different.” They further clarify that “the goal of this work is not to perfectly emulate a cellular automata (CA) in ASI.” This admission confirms my original position that the work presents a “glider-like” behavior in ASI rather than a true implementation of CA gliders. This difference is important when considering the actual contribution of this work. The title “Glider in Artificial Spin Ice” and numerous references to cellular automata throughout the manuscript remain potentially misleading. A more accurate title would be something like “Snake with Glider-like Behavior in Artificial Spin Ice,” which would better reflect the actual nature of the discovery.

While the term "glider" may be used strictly to refer to a specific structure in the context of GoL, we use the term more broadly to describe any moving structure. Our use of the term agrees with the literature, where the term "glider" is not restricted to GoL, but used in a more general sense to describe any moving structure in a CA. See for example the definition of "glider" in 9.2.3 of Adamatzky (2010), and the use of the term "glider" in a CA system defined on the Penrose tiling(Goucher, 2012).

In the introduction of the revised manuscript, we explicitly state how we use the term:

"The concept of a glider can be generalised beyond GoL, to any structure that can move through a substrate while maintaining its form, i.e., translation."

in accordance with Adamatzky (2010) (9.2.3). We have added this citation to the Introduction. We clarify further that:

"Our goal is not to implement GoL in ASI, but to search for configurations where glider-like behaviour can be found."

With these clarifications in place, it is clearly stated that we are not implementing GoL in ASI. Instead, we take inspiration from the behaviour of GoL gliders, and search for similar phenomena in ASI. As we argue in the paper, glider-like phenomena is a crucial step towards integration of information transmission and storage in ASI at the substrate level.

I acknowledge that the authors have discovered a novel and interesting collective motion pattern in ASI that exhibits some properties similar to gliders. This finding is different from previously reported phenomena such as monopole dynamics and represents a new contribution to the field. However, I maintain that the significance of this work is substantially lower than what would be expected for a true implementation of cellular automata gliders in ASI. The newly added sections on robustness and computational capabilities, while informative, do not fundamentally alter this assessment.

While we agree that a GoL implementation in ASI would be a significant achievement, we argue that there are clear similarities between CAs and ASI systems. Given these similarities, we think it is reasonable to explore computational properties in ASI within a CA framework, e.g., gliders for information transmission and storage.

Previously, the reviewer argued that the ASI differ fundamentally from a CA because the "local rules of individual units are not clearly defined". In response, in the revised manuscript, we added a rule table (Fig. 3) which shows how the head and tail of the snake moves, deterministically, under the applied clock fields. It remains unclear what additional criteria would be required for "a true implementation of cellular automata gliders".

Within the CA community, the definition of a CA is quite broad(Codd, 1968). A CA consists of a lattice of cells, where each cell can be in a finite number of states, and the state of each cell is influenced by the cells in a finite neighbourhood. Note that, under this definition, CA cells are not necessarily square and on a grid, can have a non-binary state, and have interactions beyond the Moore neighbourhood. There are even CA variants where space and time is continuous(Chan, 2019).

Similarly, an ASI consists of 2D lattice of magnets, where each magnet can be in two possible macrostates, and the state of each magnet is primarily influenced by the magnets in its immediate neighbourhood. Clearly, there are also differences between a CA and ASI, e.g., micromagnetic phenomena in ASI may play an important role in the emergent behaviour, and is not captured by the CA abstraction. However, we argue the aforementioned similarities between the two systems warrants the analogy used in the paper. This view is also supported by the other reviewers. As such, we maintain that our snake glider phenomena is as much of a "true glider" as any other CA glider. It just happens to be realized in a physical substrate rather than an abstract simulation.

Feedback from Reviewer #2:

I thank the authors for their carefully considered responses and updates to the manuscript. They have adequately addressed my concerns and those of other reviewers. The additional sections better place this work in the context of computing.

We thank the reviewer for the positive response to our revised manuscript.

I just have one small addition. At the bottom of page 6 in the "Experimental demonstration" section the authors write:

"The initial snake state is written onto a polarised ASI using a magnetic force microscopy (MFM) tip."

There are two papers that describe this kind of tip writing and should be cited in the text. The authors should also state whether this was done with or without an applied magnetic field. The papers are:

Yong-Lei Wang et al. ,Rewritable artificial magnetic charge ice.Science352,962-966(2016).DOI:10.1126/science.aad8037

Gartside, J.C., Arroyo, D.M., Burn, D.M. et al. Realization of ground state in artificial kagome spin ice via topological defect-driven magnetic writing. Nature Nanotech 13, 53–58 (2018). <https://doi.org/10.1038/s41565-017-0002-1>

After including these references I recommend publication.

We thank the reviewer for these suggestions, and have added the two citations and clarified the application of a bias field in the Results section. The details of this field are specified in the Methods section.

Feedback from Reviewer #3:

I have read with interest the revised version of the manuscript by Penty et al., and I commend the authors for the significant improvements made. The inclusion of the "Robustness" and "Towards Computing" sections enhances both the scientific depth and relevance of the work. The revised manuscript now more clearly articulates the rationale for investigating the "snake glider" as a platform for information transmission, storage, and transformation in artificial spin ice (ASI).

One of the central contributions of this work is the demonstration that controlled, glider-like motion is possible in ASI under astroid clocking, an achievement that may hold substantial implications for information processing in nanomagnetic systems. I find the development of a global clocking protocol that enables deterministic movement of a domain-like excitation ("the snake") particularly compelling—especially given its robustness to certain levels of disorder and its ability to self-correct after moderate defects.

We thank the reviewer for their positive feedback on our revised manuscript.

That said, a number of conceptual and technical points merit further clarification or expansion.

Major Points: While the paper discusses the computing potential of the snake glider and its qualitative advantages over earlier ASI-based approaches, I still find that it falls short in substantiating these claims through quantitative benchmarks. The "Towards Computing" section does a better job than the previous version in highlighting how the snake could serve as a memory or logic element. However, if the authors aim to assert that the snake glider can function as an effective substrate

Figure 1: Memory capacity curves for 50×50 pinwheel ASI utilizing snakes for information transmission.

for reservoir computing (RC), it would strengthen the manuscript considerably to include results on an established RC task—such as predicting the Mackey–Glass time series at various horizons ($t+10$, $t+20$, etc.) or computing memory capacity curves as in Jaeger (2001).

As it stands, the paper gestures toward the potential for RC but lacks a direct test of this hypothesis. This is particularly important given the relatively bold claims made about integrating memory, transmission, and logic in one substrate.

Indeed, a natural next step is to exploit the snake glider for computing, e.g., within a RC framework, integrating the transmission, storage and transformation in one substrate. However, this is outside the scope of the paper, where we focus on the snake as a foundation for information transmission and storage in a ASI computing system. We argue that such a foundation is not only relevant for RC, but for any computing paradigm exploiting the rich behaviour of ASI.

That being said, in the new “Towards Computing” section, we demonstrate how snakes can be exploited to realize a shift register in the 2D lattice of the ASI (Supplementary Movies 8 & 9). A shift register of length N will have a memory capacity $MC = N - 1$, as each input u_t is present in the register for N time steps (until it is shifted out when reaching the end of the register). The memory capacity curve for a shift register does not provide much insight: a step function which is 1.0 for delays $k = [1, N - 1]$, then drops to zero for $k \geq N$.

Figure 1 shows the MC curve for the 50×50 pinwheel ASI used in the manuscript, where a random stream of bits are encoded as snakes which are shifted into the ASI from the left (see Supplementary Movie 8). This would be the binary equivalent of Jaeger’s MC benchmark (Jaeger, 2001). As can be seen, the R^2 curve is 1.0 (perfect memory) for delays $k = [1, 23]$, then zero for $k > 23$. We have not included this plot in the manuscript, since it provides very little information.

Since a minimum snake length of two is needed to reliably encode a bit, the length of our snake shift register is $N = \frac{W-2}{2}$ where W is the width of the pinwheel ASI, and we subtract two layers of buffer magnets at the rightmost edge. Correspondingly, the resulting $MC = \frac{W-2}{2} - 1$. For the 50×50 pinwheel ASI used in the manuscript, we obtain $MC = 23$. We have added these details to the “Towards Computing” and “Methods” sections of the manuscript.

I appreciate the authors’ inclusion of references to Gartside et al. (2022) and Stenning et al. (2024), who showed that the trade-off between memory and nonlinearity in ASI can be mitigated via multilayer or multi-modal architectures. However, I still

feel this prior work should be more explicitly contrasted with the current approach. For example: Is the snake glider operating in a different part of the design space? Does it offer advantages in energy efficiency, reconfigurability, or scalability that are not achievable through spin-wave-based RC?

Indeed, it is possible to compensate for a lack of memory capacity in the reservoir substrate by employing multilayer architectures or delay lines. Note that multilayer architectures can implicitly add memory as data is shifted between layers. However, we argue that such architectures are costly in terms of the required circuitry and the need for constantly reading out reservoir state between layers. In any computer, external memory operations (reading or writing state) are several orders of magnitude more energy demanding than operations that can be kept on the same substrate (Dally et al., 2020). Hence, there is a huge efficiency advantage if the substrate can be made to support memory directly, instead of relying on external support circuitry. That is exactly what we have shown with the snakes. In terms of scalability, we have shown how MC scales linearly with size (width) of the ASI, when using snakes as information carriers. Such scalability has not been demonstrated for other ASI RC systems. We have added the reference to Dally et al. (2020) to the introduction to clarify the significant overhead of external memories. Additionally, we have highlighted the linear scalability of snake-based MC in the Discussion.

Perhaps the most intriguing aspect of the manuscript is the ability to control the glider’s motion deterministically through clock fields, which elevates this from a curiosity to a potential building block. This level of control is a nontrivial step forward and deserves further emphasis. It would be beneficial to explain whether this level of control could be extended to more complex logic operations or routing schemes in a 2D ASI network.

Good point, we have elaborated on this aspect in the discussion, where we emphasize the deterministic / local control / global clock fields property of the technique. Inspired by the reviewer’s comments on more complex logic operations or routing schemes, we have added a sentence to the discussion regarding the potential use case of the snake in a reconfigurable magnetic device.

The snake collisions presented in the new “Towards Computing” section is a first step in this direction, where we demonstrate that two horizontally moving snakes may interact vertically, thereby also making use the second dimension of ASI. Future work is sure to explore and extend these possibilities.

In traditional RC or neuromorphic paradigms, learning is often implemented through feedback and adaptation of internal weights. In contrast, the snake glider system relies on structure-driven dynamics governed by externally applied fields. It would be helpful if the authors could clarify whether, and how, feedback-driven learning might be realized in this framework—for instance, via adaptive modulation of clock field strength, snake initialization, or environmental inputs. This would help bridge the gap between biological inspiration and functional implementation.

It is true that in many neuromorphic paradigms, learning is realized through adaptation of internal weights. However, this is not the case in RC, where the internal reservoir weights remain fixed. This is also the case in most, if not all, ASI systems, where the magnet coupling is dictated by the geometry (arrangement of magnets) and hence the “internal weights” are fixed at fabrication time. It is, however, possible to adaptively alter behaviour through external fields, as we have demonstrated in the “Towards Computing” section. There, we show how snake-snake interactions can be tuned by changing the strength of the clock fields. There is also adaptation as part of the evolutionary algorithm feedback loop, in the sense that the algorithm “learns” an initial state and field strengths that produces some desired behaviour. The methodology

can, of course, be extended towards other functional building blocks suitable for solving specific tasks. However, such a study is outside the scope of the paper, where we focus on the snake as a foundation for information transmission and storage in ASI.

Minor Comments: The use of the term "glider" is appropriate and well-justified through analogy with cellular automata, but I recommend being careful not to overstate the equivalence. The emergent dynamics are clearly more constrained than in Game of Life, where gliders can interact in richer ways. That said, the glider analogy remains helpful and illustrative.

We appreciate the comment on the suitability of the "glider" analogy. Indeed the systems are not equivalent, as we have emphasized in the introduction. Still, it is not clear in what aspects the GoL gliders display more rich interactions, compared to the snakes. A rich variety of snake-snake interactions are demonstrated in the new "Towards Computing" section. Perhaps the reviewer is alluding to the four possible directions of travel, compared to the two directions of the snakes? Indeed, combined horizontal/vertical movement is a crucial fundamental step which must be addressed in the future. We also have to acknowledge the fact that GoL has been studied extensively for decades. The ability to study ASI systems as discrete dynamical systems was developed only very recently with astroid clocking. In the future, we are sure to discover a wide variety of CA-like magnetic patterns.

Figure 8, showing collision dynamics, is one of the most compelling demonstrations in the paper. If possible, it would be valuable to elaborate on whether these transformations can be harnessed in a programmable fashion, or if they remain stochastic due to disorder.

As mentioned in the text, the snake collisions in Figure 8 were the result of simulations without disorder. Given the results presented in the Robustness section, it is highly likely that disorder would alter the outcome of many of these collisions. Exactly how disorder would affect collisions could be an interesting topic for a future study. Given the self-correcting behaviour displayed by the snakes experimentally (Robustness section), we are hopeful that reliable transformations are obtainable in the presence of disorder.

The revised introduction is clearer, though still somewhat long. The discussion of prior limitations in ASI computing should come earlier, to better motivate the problem the authors address.

A discussion of the prior limitations in ASI computing already appears in the second paragraph of the introduction. We feel it is difficult to include this already in the first paragraph, where we introduce ASI at a conceptual level for the unfamiliar readers.

Recommendation: While I believe this work represents a meaningful and creative step in ASI-based information processing, I still have some reservations about its readiness for Nature Communications. In particular, I would expect either (1) a demonstration of a concrete computational benchmark using the snake glider, or (2) more extensive validation of its scalability and applicability to real-world neuromorphic computing tasks.

That said, this is clearly a high-quality contribution and one that would be well-suited for a specialized or emerging journal in the field, such as Communications Physics, Scientific Reports, or Advanced Intelligent Systems. Should the authors include a proof-of-principle RC task, I would be inclined to reconsider the suitability for Nature Communications.

With the added clarifications on the memory capacity benchmark (Jaeger, 2001) of the snake-based shift register, we argue that a concrete computational benchmark has been demonstrated. As such we believe the manuscript now meets the reviewer’s criteria for acceptance.

Feedback from Reviewer #5:

The authors have made substantial efforts to improve their manuscript and it is improved as a result. I still find the experimental results slightly disappointing - it is clear that disorder in the array is greatly inhibiting the ability to reliably transmit gliders through the array. However, the additional measurements and simulations of the effects of disorder provided clear explanations and indications of the levels of disorder required to produce more reliable motion.

Yes, disorder is unfortunately detrimental to the snake glider. It is likely that disorder can be improved significantly by optimizing the fabrication details. In the future, we hope to find other glider structures, beyond the simplest snake, which are more resilient to disorder.

More generally the revisions have made the manuscript more thoughtful on the implications of their discoveries, and I found it interesting to read. I remain convinced it is a step forward for the field with interesting implications for computing research.

Overall, I think this now merits publication. It would be great if more reliable motion could be demonstrated, but I wouldn’t want to inhibit this being released to the community by insisting on this.

Thank you for the recommendation.

References

- Adamatzky, A., editor (2010). *Game of Life Cellular Automata*. Springer London, London.
- Chan, B. W.-C. (2019). Lenia - Biology of Artificial Life. *Complex Systems*, 28(3):251–286.
- Codd, E. F. (1968). *Cellular Automata*. Academic Press, New York.
- Dally, W. J., Turakhia, Y., and Han, S. (2020). Domain-specific hardware accelerators. *Commun. ACM*, 63(7):48–57.
- Gartside, J. C., Arroo, D. M., Burn, D. M., Bemmer, V. L., Moskalenko, A., Cohen, L. F., and Branford, W. R. (2018). Realization of ground state in artificial kagome spin ice via topological defect-driven magnetic writing. *Nature Nanotechnology*, 13(1):53–58.
- Goucher, A. P. (2012). Gliders in Cellular Automata on Penrose Tilings. *Journal of Cellular Automata*, 7(5/6):385–392.
- Jaeger, H. (2001). Short term memory in echo state networks.
- Stenning, K. D., Gartside, J. C., Manneschi, L., Cheung, C. T. S., Chen, T., Vanstone, A., Love, J., Holder, H., Caravelli, F., Kurebayashi, H., Everschor-Sitte, K., Vasilaki, E., and Branford, W. R. (2024). Neuromorphic overparameterisation and few-shot learning in multilayer physical neural networks. *Nature Communications*, 15(1):7377.
- Wang, Y.-L., Xiao, Z.-L., Snezhko, A., Xu, J., Ocola, L. E., Divan, R., Pearson, J. E., Crabtree, G. W., and Kwok, W.-K. (2016). Rewritable artificial magnetic charge ice. *Science*, 352(6288):962–966.

The authors have demonstrated through simulation and experiment how to create cellular automata (CA) in a pinwheel artificial spin ice (ASI). The authors used an evolutionary algorithm to discover an equivalent CA to the “glider” in Conway’s Game of Life which they call a “snake”. They demonstrate the linear movement of a “snake” via a simple “astroid clocking” field protocol. They explain how the dipolar field of nearest neighbours stabilise and destabilise the ends of the “snake” to allow it to transmit through the lattice in a stable way. The authors argue that this will be of use to ASI based computing which currently lacks short term memory. I think this is fantastic work and recommend publication after some minor corrections and a suggested expansion of the discussion.

I think the use of time-evolution (dynamics) could be misinterpreted as magnon or spin-waves by some. I suggest (reversal-dynamics) or something similar.

After carefully following the authors analysis of the nearest neighbour’s dipolar field effect on the snake I think the effective snake looks something like this:

Should the snake be considered as only the reversed magnets (red box) or the surrounding nearest neighbours also (black box)? I think an indication of an effective boundary for the snake will be useful for the reader.

It is a shame that the “pointy” end and the tail is with the “forked” end. Intuitively I think of a snake’s tail as being pointy and having a forked tongue for the head. The authors should either pick different nomenclature or carefully describe what they mean by pointy and forked to avoid reader confusion. Perhaps using heads and tails of an arrow would be better, but this is just a suggestion.

Are the authors able to elaborate on how they would nucleate a functional snake? The authors point out that the smallest functional snake has length two where there are two magnets from each sublattice. Is it possible to create the snake structure from the above left? Here the tail and head are

effectively the same. But if it is possible to do this you should then be able to change the direction of a snake shrinking to the left structure and then rotating the clock fields 90 degrees. Perhaps a different clock sequence is required to transform between the two states below. I think if the authors can demonstrate this functionality, it would vastly increase the impact of the paper since now you are making use of the 2D structure more fully.

The authors could cite some papers on how initialisation of structures could be achieved in practice. For example:

Nature Nanotechnology volume 13, pages 53–58 (2018)

Cell Reports Physical Science, Volume 4, Issue 3, 15 March 2023, 101291

Communications Physics volume 3, Article number: 219 (2020)

The authors mention that the nearest neighbour dipolar fields push the switching fields outside the switching asteroid. Is there a way to indicate the effect on an asteroid plot for the cases shown in fig 4. This could be included as supplementary information.

The following is probably beyond the scope of the current study but I think it would be useful to include some answers to these as possible future directions. In the game of life, the rules are set and the complex behaviour emerges from the initialised state. The authors use an evolutionary algorithm that varies both initial state and the clock fields. For a fixed clock field do the authors find other cellular automata? Ideally you would want the same clock algorithm to apply to get the complex computations from the initialised state alone. If the clock fields used here can apply to other structures this would be very exciting.

The authors also neglect structures with frozen or oscillatory behaviour but these can be instrumental in generating snakes. For example, the Gosper glider gun in GoL consists of static and oscillatory elements to produce a steady stream of gliders. Can the EA's fitness function be modified to reward an oscillating structure that also generates snakes indefinitely?

Referee Report for “Snakes in the Plane: Controllable gliders in a nanomagnetic material”

I read with great interest the manuscript by Penty et al. I was intrigued by the title at first. The manuscript is undoubtedly intriguing as a whole, and contains several valuable contributions, particularly in exploring the role of "snake gliders" in pinwheel, a type of artificial spin ice lattice. The work builds on prior knowledge mostly developed previously by a subset of these authors, and by creatively applying the concept of gliders to the ASI system and provides a comprehensive experimental and theoretical analysis. However, as much as the manuscript is well-constructed, it leaves the reader wanting more—there is a certain lack of impact or a "punch" that would elevate this work from a solid contribution to a standout paper. Below, I elaborate on key areas where improvements could be made to enhance the clarity, accessibility, and scientific significance of the manuscript. I also attach my comments directly to the text below. My report below should be read within the context of those comments.

First, while the concept of the snake glider is well-motivated by parallels to cellular automata and the Game of Life, the broader significance of these findings for computation is not fully developed. The idea of leveraging ASI for computation has been explored before, including these authors, but this paper seems to leave the potential applications of the snake glider in neuromorphic or unconventional computing somewhat underexplored. While the authors suggest that the glider can transmit and store information, there is a missed opportunity to more explicitly position this as a breakthrough in integrating memory and information transfer in ASI systems. For instance, can this glider outperform other proposed mechanisms for memory in ASI? Is it scalable or robust enough to serve as a basis for real-world implementations? These points need clearer articulation. I see value in this contribution, but at the same time it leaves me wondering how to use it.

Second, the authors describe the evolutionary algorithm (EA) used to discover the snake glider, but the explanation lacks sufficient detail to be convincing to the reader. The mechanism of the EA, the fitness function, and the parameter optimization process are only briefly described and would benefit from greater transparency instead of being buried in the Methods. For example, what were the key constraints or challenges in finding a stable glider, and how were these overcome? Including a more thorough discussion of the EA's success or failure in identifying other gliders would provide additional depth and allow the reader to understand whether this method could be generalized for other magnetic systems.

The manuscript would also benefit from a more direct comparison with prior work in the field, at least at the level of context. While there are scattered references to studies on domain growth and magnetic monopoles, these feel somewhat disconnected from the main narrative of computing. A stronger engagement with foundational work in ASI and neuromorphic computing would help clarify what differentiates this contribution. I did add some references, feel free to pick from them.

A related point is that the introduction, while thorough in describing the basic principles of ASI and cellular automata, feels somewhat long-winded and does not lead to a compelling research

question. The problem of information storage and transmission in ASI is well-known, but the authors miss the chance to highlight the gap their work is addressing. The introduction could be streamlined and restructured to foreground the novelty of the snake glider concept and its broader implications.

Furthermore, while the experimental validation of the snake glider is impressive, it remains unclear whether the demonstrated robustness is sufficient for practical applications. The discussion notes that fabrication defects and coercive field variations can break down the glider's motion, but does not provide a sense of how frequently this occurs or how it could be mitigated. Similarly, while the authors show that the snake is capable of bi-directional movement, the potential for interacting snakes or more complex dynamics is not explored. This feels like a missed opportunity to extend the work beyond the relatively simple demonstrations shown in the manuscript.

Lastly, while the figures and visualizations are generally clear and well-presented, there is a lack of intuitive explanation for some of the observed phenomena. For example, the dipolar interactions that govern the snake's movement are described in detail, but the connection between these interactions and the macroscopic glider motion is not made intuitive. A more conceptual or diagrammatic explanation of why the snake moves and why its head and tail behave differently would be a welcome addition.

In summary, while this work provides an innovative demonstration of gliders in ASI and is technically sound, it does not quite achieve its full potential. The manuscript is missing the critical impact or conceptual breakthrough that would make it truly memorable. By strengthening the connection to prior work, expanding the discussion of potential applications, and better contextualizing the importance of the findings, the authors could elevate this work to a higher level. For now, the paper reads as an interesting but somewhat niche contribution, rather than a transformative piece of research. I hope the authors will take these suggestions as constructive feedback to refine their message and broaden the appeal of their work.

There is a history of works similar to this one being published in Nature Communication. However, at this stage I honestly think that this is subpar compared to those - although I liked this paper and its innovative way of using/bring CA ideas into this field. This is definitely a good attempt/achievement towards this, I am just not sure if it is quite there to deserve a Nature Communication publication. I could see this published in Comm in Phys after a minor revision.